# A Gaussian Mixture Method for Specific Differential Phase Retrieval at X-band Frequency

Guang Wen[1], Neil I. Fox[1,*], and Patrick S. Market[1]

[1]School of Natural Resources, University of Missouri, 332 ABNR Building, Columbia, Missouri, USA, 65201

**Correspondence:** Neil I. Fox (foxn@missouri.edu)

**Abstract.** Specific differential phase $K_{dp}$ is one of the most important polarimetric radar variables, but the variance $\sigma^2(K_{dp})$, regarding the errors in the calculation of the range derivative of differential phase shift $\Phi_{dp}$, is not well characterized due to the lack of a data generation model. This paper presents a probabilistic method based on Gaussian mixture model for $K_{dp}$ estimation at X-band frequency. The Gaussian mixture method can not only estimate the expected values of $K_{dp}$ by differentiating the expected values of $\Phi_{dp}$, but also obtain $\sigma^2(K_{dp})$ from the product of the square of the first derivative of $K_{dp}$ and the variance of $\Phi_{dp}$. Additionally, ambiguous phase and backscattering differential phase shift are corrected via the mixture model. The method is qualitatively evaluated with a convective event of a bow echo observed by the X-band dual-polarization radar in the University of Missouri. It is concluded that $K_{dp}$ estimates are highly consistent with the gradients of $\Phi_{dp}$ in the leading edge of the bow echo, and large $\sigma^2(K_{dp})$ occurs with high variation of $K_{dp}$. Furthermore, the performance is quantitatively assessed by two-year radar-gauge data, and the results are compared to linear regression model. It is clear that $K_{dp}$-based rain amounts have good agreement with the rain gauge data, while the Gaussian mixture method gives improvements over linear regression model, particularly for far ranges.

## 1 Introduction

Apart from radar reflectivity ($Z_H$) and differential reflectivity ($Z_{DR}$), polarimetric radars also obtain differential phase shift ($\Phi_{dp}$) to reflect the forward scattering property of hydrometeor scatterers (Seliga and Bringi, 1978; Sachidananda and Zrnić, 1986). Its range derivative, also called specific differential phase ($K_{dp}$), has some advantages over $Z_H$ and $Z_{DR}$ (Zrnić and Ryzhkov, 1996), including insensitivity to attenuation, clutter, partial beam blockage and radar absolute calibration. The specific differential phase has played a key role in various meteorological applications—such as hydrometeor classification (Lim et al., 2005; Park et al., 2009), raindrop size distribution retrieval (Bringi et al., 2002; Williams et al., 2014) and quantitative precipitation estimation (Ryzhkov et al., 2005; Cifelli et al., 2011)—since $K_{dp}$ is a phase variable independent of $Z_H$ and $Z_{DR}$ and almost linearly proportional to rain rate.

A linear regression model has been developed to derive $K_{dp}$ from the slope of the range profile of $\Phi_{dp}$ measured by the polarimetric radars. In Hubbert et al. (1993), $\Phi_{dp}$ is first processed by a light filter that attenuates the $\Phi_{dp}$ magnitudes within a scale of 375 m by 10 dB, and then heavily smoothed in 1.5 km by 10 dB. An iterative filtering technique is used for eliminating non-zero backscattering differential phase shift (Hubbert and Bringi, 1995). The filtered $\Phi_{dp}$ measurements are finally fitted into a first-order polynomial to estimate the $\Phi_{dp}$ slope in a given window. Liu et al. (1993) supply the accuracy of mean $K_{dp}$ as $\pm 0.25$ deg km$^{-1}$ using 128 pulses, while Aydin et al. (1995) indicate that the accuracy is within $\pm 0.5$ deg km$^{-1}$ for a heavy rainfall event using 64 pulses. On the other hand, Ryzhkov and Zrnić (1996) produce two kinds of $K_{dp}$ for S-band radars: one is obtained over 16 range gates (2.4 km) for $Z_H \leq 40$ dBZ, and the other is produced over 48 gates (7.2 km) for $Z_H > 40$ dBZ. Negative $K_{dp}$ is incorporated into the rain rate algorithm to avoid bias in low rain rate. The analyses of 15 storms show that the standard error of $K_{dp}$ is 0.04~0.10 deg km$^{-1}$ for heavily filtered $K_{dp}$ and 0.12~0.30 deg km$^{-1}$ for lightly filtered $K_{dp}$ using either 128 or 64 pulses. Vulpiani et al. (2012) develop a multi-step moving window approach based on the linear regression model to handle the $\Phi_{dp}$ folding and other ambiguous data. This approach is applicable for the complex terrain, but still valid for various topographical environments.

X-band dual-polarization radars have drawn increasing attention in the radar meteorology community in recent years on account of low cost, fine resolution and high sensitivity to light precipitation (Chandrasekar et al., 2012; Lim et al., 2013; Berne and Krajewski, 2013; Kalogiros et al., 2014; Oue et al., 2016). In the literature, X-band algorithms have been proposed for $K_{dp}$ estimation. For example, the linear regression method is adapted for the X-band radar data, and used to retrieve rainfall (Matrosov et al., 2006). The ambiguous $\Phi_{dp}$ is naturally corrected by examining the complex values of the range profiles of $\Phi_{dp}$ exponentials, and $K_{dp}$ is then estimated by a regularization framework based on a cubic spline smoothing (Wang and Chandrasekar, 2009). In this method, the bias and variance are adjustable through the smoothing parameter, giving high spatial resolutions of $K_{dp}$ estimates. Moreover, algorithms of linear programming (Giangrande et al., 2013) and Kalman filter (Schneebeli et al., 2014) have also been applied to the $K_{dp}$ estimation, yielding good performance for rainfalls and snowfalls. It is noticeable that the Kalman filter method minimizes the Gaussian error function to obtain the mean profile of $K_{dp}$. It gives a significant improvement on the $K_{dp}$ mean, particularly in the small-scale structure with high peaks. In addition, the $\Phi_{dp}$ measurements at X-band frequency are affected by backscattering differential phase shift $\delta_{co}$. The constraints of $K_{dp} - Z_H - Z_{DR}$ and $\delta_{co} - Z_{DR}$ can be used to improve the estimation of $K_{dp}$ and $\delta_{co}$ (Otto and Russchenberg, 2011; Reinoso-Rondinel et al., 2018), although these constraints are only valid in the rain regime.

The recent algorithms are focused on the improvement of estimating the mean $K_{dp}$, whereas its variance $\left(\sigma^2(K_{dp})\right)$ is not well characterized due to the lack of a data generation model. The $K_{dp}$ variance is often inherited from the $\Phi_{dp}$ variance $\left(\sigma^2(\Phi_{dp})\right)$ leading to large relative errors for low $K_{dp}$ with a fixed path length. As noted by Gorgucci et al. (1999), the $K_{dp}$ estimated by the linear regression has large errors in the nonuniform rain media, while the errors increase when the radar reflectivity presents large gradients in dimensions. In this study, we propose a probabilistic method based on Gaussian mixture model for $K_{dp}$ estimation at X-band frequency. The Gaussian mixture method can not only estimate the expected values of $K_{dp}$ by differentiating the conditional expectation of $\Phi_{dp}$, but also yield $\sigma^2(K_{dp})$ by regarding the errors in the calculation of the first derivative of $\Phi_{dp}$. It is found that $\sigma^2(K_{dp})$ is closely related to the square of the first derivative of $K_{dp}$ and $\sigma^2(\Phi_{dp})$,

while large $\sigma^2(K_{dp})$ is associated with high variation of $K_{dp}$ estimates. When compared to the existing methods, our method considers the joint probability density function of the data as the non-linear Gaussian mixture, leading to better performance for the multimodal data. Since the $K_{dp}$ variance is non-constant, it leads to the variability in the $K_{dp}$ error characteristics. We can then use the $K_{dp}$ variance to calculate the variances of $Z_H$ and $Z_{DR}$ via the attenuation correction, and the variance of rain rate via the $R$-$K_{dp}$ relation. These variances are useful for studying the propagation of uncertainty in the weather model and the stream flow trends in the hydrological model.

The paper is organized as follows. Section 2 provides background information about $K_{dp}$ and the Gaussian mixture model. Section 3 describes the radar and gauge data. Section 4 presents the methodology. We first remove the residual clutter using data masks (section 4.1), and then derive the joint probability density function to estimate the expected value of $\Phi_{dp}$ and $\sigma^2(\Phi_{dp})$ (section 4.2). Next, we correct the ambiguous phase and $\delta_{co}$ via the mixture model (section 4.3). Last, we calculate the expected value and variance of $K_{dp}$ (section 4.4), and improve the $K_{dp}$ profile by reducing $\sigma^2(K_{dp})$ (section 4.5). To evaluate the algorithm, section 5 gives a case study and a comparison between radar and gauge. Section 6 summarizes the paper.

## 2    Background

The specific differential phase is the first derivative of differential phase shift $\Phi_{dp}$ along the radar range, giving a way to estimate $K_{dp}$ by radar measurement of $\Phi_{dp}$. Furthermore, the probability density function of $\Phi_{dp}$ can be modelled as a Gaussian mixture, which is often obtained via an expectation-maximization (EM) approach. The mean and variance of the Gaussian mixture may lead to the improvement of the $K_{dp}$ estimation.

In this section, we introduce the physical interpretation of $K_{dp}$ and the regression model for estimating $K_{dp}$. Since the Gaussian mixture is adopted as the data generation model, we also give a brief description of mathematical definition of the Gaussian mixture model and the EM approach.

### 2.1    Specific differential phase ($K_{dp}$)

For linear polarization, $K_{dp}$ is proportional to the integral of the raindrop size distribution and the real part of the difference of forward scattering amplitudes at orthogonal polarizations. It is mathematically formulated as

$$K_{dp} = \frac{0.18\lambda}{\pi} \int_0^\infty N(D) \cdot \Re\left[f_{hh}(0,D) - f_{vv}(0,D)\right] dD \quad (\text{deg km}^{-1}), \tag{1}$$

where $\lambda$ is radar wavelength in millimeters, $D$ is raindrop size in millimeters, $N(D)$ is size spectrum in m$^{-3}$mm$^{-1}$, $f_{hh,vv}(0,D)$ is forward scattering amplitudes at horizontal and vertical polarizations, respectively.

By considering the Rayleigh-Gans scattering from identical and horizontally-oriented oblate spheroids, such as raindrops, the forward scattering amplitudes are proportional to the inverse square of radar wavelength, i.e., $f_{hh,vv}(0,D) \propto 1/\lambda^2$, leading to the fact that $K_{dp}$ is inversely proportional to radar wavelength, i.e., $K_{dp} \propto 1/\lambda$. Therefore, the values of $K_{dp}$ at X-band

are often larger than that at S-band by a factor of 3, indicating that X-band radar can provide better $K_{dp}$ data than S-band radar when retrieving the rainfall rate. The conclusion is still valid even if the Mie effect is taken into account (Bringi and Chandrasekar, 2001; Chandrasekar et al., 2006).

However, $K_{dp}$ cannot be detected by polarimetric radar directly, whereas its integral $\Phi_{dp}$ is measurable. Hence, $K_{dp}$ can be estimated as the range derivative of the profile of $\Phi_{dp}$, i.e., $K_{dp} = \frac{\Delta\Phi_{dp}}{2\Delta r}$, where $r$ is the radar range in kilometers. An alternative approach to estimating $K_{dp}$ is to apply a regression fit to the profile of $\Phi_{dp}$, and the first order polynomial is usually considered as the fitting function (Balakrishnan and Zrnić, 1990; Ryzhkov and Zrnić, 1995). Subsequently, if the $\Phi_{dp}$ measurements are equally spaced in range by $\Delta r$, $K_{dp}$ is then estimated by

$$K_{dp} = \frac{\sum_{i=1}^{n} \Phi_{dp}(r_i)\left[6i - 3(n+1)\Delta r\right]}{n(n-1)(n+1)\Delta r^2}, \tag{2}$$

where $n$ is the number of gates. Equation (2) shows that the accuracy of $K_{dp}$ estimates is determined by the number of gates ($n$), and the accuracy of $\Phi_{dp}$. By assuming $\sigma^2(\Phi_{dp})$ is relatively stable for all gates along a ray and noting that $\Phi_{dp}(r_i)$ is the only variable in Eq. (2), $\sigma^2(K_{dp})$ is formulated as

$$\sigma^2(K_{dp}) = \frac{3\sigma^2(\Phi_{dp})}{\Delta r^2\left[n(n-1)(n+1)\right]}. \tag{3}$$

In Eq. (3), $\sigma^2(K_{dp})$ is proportional to $\sigma^2(\Phi_{dp})$, which is related to the spectrum width, cross-correlation coefficient, and the dwell time (Sachidananda and Zrnić, 1986; Hubbert et al., 1993), and inversely proportional to $n^3$. This method has been widely used in the existing radar system (Cifelli et al., 2018; Chandrasekar et al., 2018; Chen et al., 2017c, b). The details of the regression-based estimation of $K_{dp}$ are given in Bringi and Chandrasekar (2001) and Appendix A.

Moreover, it is notable that the backscattering phase shift is not negligible at X-band, thus the total propagation phase shift ($\Psi_{dp}$) consists of $\Phi_{dp}$ and the backscattering differential phase, $\delta_{co}$, i.e., $\Psi_{dp} = \Phi_{dp} + \delta_{co}$. The backscattering phase shift is often showed as a sudden jump over one or few range gates in a monotonically increasing $\Psi_{dp}$ profile of rain (May et al., 1999a), with a value much larger than standard deviation $\sigma(\Psi_{dp})$. The presence of $\delta_{co}$ over a small number of consecutive gates can be eliminated by a simple filter (Hubbert and Bringi, 1995).

The specific differential phase is a unique polarimetric variable in terms of statistical errors in the rain rate estimation, since it is the range derivative of the phase measurement $\Phi_{dp}$. The errors in the calculation of the first derivative also needs to be taken into account. In this study, we consider a Gaussian mixture as the data generation model, which plays an important role in the estimation of $K_{dp}$ and $\sigma^2(K_{dp})$.

## 2.2 Gaussian mixture model

The Gaussian mixture is a statistical model for data probability density estimation, assuming that the data points are generated by a mixture of a finite number of Gaussian distributions associated with their weights (McLachlan and Peel, 2000; Sung, 2004). Intuitively, it is used to model the multimodal data, with each Gaussian component corresponding to a subpopulation of

the data. The mathematical formulation is given as

$$f(z) = \sum_{i=1}^{m} w_i \mathcal{N}(z; \mu_i, \Sigma_i),$$ (4)

where $m$ is the number of components in the Gaussian mixture, $w_i$ is a weight with $\sum_{i=1}^{m} w_i = 1$, and $\mathcal{N}(z; \mu_i, \Sigma_i)$ is the $i$th Gaussian distribution with mean $\mu_i$ and covariance $\Sigma_i$, i.e., $\mathcal{N}(z; \mu_i, \Sigma_i) = \left| (2\pi)^k \Sigma_i \right|^{-1/2} \exp \left[ -\frac{1}{2}(z - \mu_i)^T \Sigma_i^{-1} (z - \mu_i) \right]$,

where $k$ is the data dimension.

It is prevalent to use an Expectation–Maximization (EM) algorithm to estimate the parameters, $w$, $\mu$, and $\Sigma$, by constructing the lower bound of log-likelihood based on Jensen inequality (Dempster et al., 1977). The EM algorithm is divided into two steps, namely, an expectation (E) step and a maximization (M) step. In the E step, a degree of membership toward to the $j$th cluster is calculated, i.e.,

$$Q_j^i = p\left( y^{(i)} = j | x^{(i)}; w, \mu, \Sigma \right),$$ (5)

where $i$ is the $i$th data with a total number of $n$ data points, and $y$ is a latent variable that determines the corresponding cluster. Here, $Q$ gives a tight lower bound for the log-likelihood, equivalent to maximizing the expectation. In the M step, the exact form of the lower bound based on Jensen inequality is expressed as

$$\mathcal{L}(w, \mu, \Sigma) = \sum_i \sum_j Q_j^i \log \frac{\exp \left[ -\frac{1}{2}(x^i - \mu)^T \Sigma^{-1}(x^i - \mu) \right] w_j}{\sqrt{|(2\pi)^k \Sigma|} Q_j^i}.$$ (6)

By maximizing the lower bound with respect to each parameter, $w_j$, $\mu_j$, and $\Sigma_j$ are updated as (Petersen and Pedersen, 2012)

$$w_j = \frac{\sum_i Q_j^i}{n},$$ (7)

$$\mu_j = \frac{\sum_i Q_j^i x^{(i)}}{\sum_i Q_j^i},$$ (8)

$$\Sigma_j = \frac{\sum_i Q_j^i (x^{(i)} - \mu_j)(x^{(i)} - \mu_j)^T}{\sum_i Q_j^i}, \text{respectively.}$$ (9)

Notably, the M step increases the log-likelihood monotonically, and the covariance retains positive definite with sufficiently
large data samples. Finally, the E step and M step are iteratively operated until the log-likelihood converges to a value with the difference between two successive steps below a certain threshold. In addition, the EM algorithm requires a specification of the number of clusters, $m$, prior to the E and M steps, and an inappropriate choice of $m$ may lead to meaningless values of the parameters. To tackle this problem, the Bayesian information criterion is often calculated to select the optimal $m$, while a Dirichlet process may also be used to model a prior probability to construct an infinite Gaussian mixture.

One of interpretations of the Gaussian mixture is to view each distribution as a cluster with a Gaussian probability density, while the individual data point is attributed to a specific cluster or a weight toward the cluster, regarded as unsupervised learning (Hastie et al., 2009). The clustering procedures based on Gaussian mixture have been applied to the identification of storm structure (Veneziano and Villani, 1996), and the particle identification at S-band (Wen et al., 2015, 2016b, 2017) and

X-band (Wen et al., 2016a) frequencies. Furthermore, the Gaussian mixture model can be extended to fit a set of unknown parameters in the prior probability of the Bayesian framework, forming a Bayesian Gaussian mixture model (Li et al., 2012). The prior is then multiplied with the known conditional probability of data given the parameters to be estimated, yielding the posterior probability with a new set of parameters. The expectation of the posterior is often used to retrieve the conditional mean of the new parameters based on least square criteria.

For the regression problem, the characteristics of the Gaussian mixture imply that the direct modeling of a regression function is very difficult. Nevertheless, the joint probability of the measurements and the estimated parameters may be modeled as a Gaussian mixture, leading to a regression function derived from the joint density model. Due to the asymptotic consistency of a Gaussian mixture model, it is capable of estimating a general density function in $\mathbb{R}^n$ in any shape (Sung, 2004). Moreover, the speed of calculating unknown parameters within a Gaussian mixture linearly depends on the number of the training data points, and the computation of the outputs is independent of the size of the training data. Consequently, regression based on a Gaussian mixture can be achieved very rapidly, compared to Gaussian process regression that grows with the data size. In addition, the Gaussian mixture can also be used to solve the regression problem with multiple dimensions, and a subset of dimensions can be selected to handle the missing data (Wen et al., 2015).

## 3 Data

As part of the Missouri Experimental Project to Stimulate Competitive Research (EPSCoR), an X-band dual-polarization radar in the University of Missouri (MZZU) was deployed at the South Farm Research Center (38.906°N, 92.269°W) in the midwest of America in the summer of 2015. The details of the radar characteristics are described in Simpson and Fox (2017). The primary objective is to provide the observations of precipitation near the surface by means of low-cost and fine-scale X-band radar, and to fill the observational gaps of the S-band radar network in Saint Louis (KLSX), Kansas City (KEAX), and Springfield (KSGF). Within the MZZU radar coverage, the Hinkson Creek located near Columbia, MO, flows through a catchment basin and eventually merges into the Missouri river, forming a typical urban watershed (Hubbart and Zell, 2013). The radar can provide timely flash flooding warning for the Hinkson Creek watershed and surrounding areas.

In this study, we analyze the data collected by the X-band MZZU dual-polarization radar. The maximum unambiguous range of the MZZU radar is 94.64 km with a resolution of 260 m in range and 1° in azimuth. During the observational periods, the radar operates in a volumetric scanning mode of nine elevations at 0.8°, 2°, 3°, 4°, 5°, 6°, 7°, 8.5°, and 10°, updated every 4 minutes. The raw radar data are organized and processed by an open-source software package called Python ARM Radar Toolkit (Py-ART: Helmus and Collis, 2016). Moreover, to validate the $K_{dp}$ estimation algorithm, we also use the data from tipping-bucket rain gauges in the Missouri Mesonet weather station network, including Bradford Farm (38.897°N, 92.218°W), Sanborn Field (38.942°N, 92.320°W), Auxvasse (39.089°N, 91.999°W), and Williamsburg (38.907°N, 91.734°W). The horizontal distances between the rain gauges and the radar center are 4.4 km, 6.0 km, 30.8 km, and 46.2 km, respectively. The first elevations at Bradford and Sanborn may be affected by ground clutter, since the radar beams are very close to the ground, with heights of 314.6 m and 336.9 m ASL, respectively, including the radar tower. Therefore, the second elevation at 2° is selected

for validation. In contrast, the first elevations at Auxvasse and Williamsburg reach about 723.8 m and 999.0 m ASL, which are less contaminated by ground clutter. Furthermore, the point measurement of rain gauge is different from the volumetric measurement of radar, imposing additional errors on the comparison between radar and gauge (Anagnostou et al., 1999). The radar-based rain rate is then derived by averaging $K_{dp}$ over three successive range gates and three successive azimuthal rays with a total of 9 values centered over each gate in order to obtain good consistency between the instruments. In addition, the rain gauges are carefully calibrated in terms of instrumentation failure, clogging, and other discrepancies between the devices (Simpson and Fox, 2017), and well documented to provide long-term data for rainfall observations.

Table 1 summarizes the characteristics of rainfalls observed at Bradford, Sanborn, Auxvasse and Williamsburg between April 2016 and June 2018. It is clear that the hourly rain amounts are dominated by light rain, with similar means of 2.0–2.1 mm at the four sites, indicating uniformly distributed rainfalls within the experimental region. On the other hand, the standard deviations of Bradford and Williamsburg are 3.5 mm and 3.7 mm, respectively, a little larger than that of 3.3 mm at Sanborn and Auxvasse. Moreover, Sanborn gives the highest hourly rain amount, the lowest total rain amount, and the second lowest duration out of the four sites, due to the effects of urban heat island (Hubbart et al., 2014). The second highest maximum hourly rain amount is recorded at Williamsburg, however, the total rain amount and duration are also the highest among the four sites, implying that convective rain is the most frequent at Williamsburg. In contrast, stratiform rain is more common at Bradford, since the gauge records the lowest maximum hourly rain amount and duration, and the second total hourly rain amount. In addition, it can be seen that Auxvasse also provides useful data for the comparisons between gauges and between radar and gauge, though the statistics are all ranked in the middle of the four sites. Overall, the rain gauge data at Bradford, Sanborn, Auxvasse and Williamsburg are representative and sufficiently large, leading to a valid dataset for testing the $K_{dp}$ and $K_{dp}$-based rain amounts.

## 4  $K_{dp}$ retrieval

As discussed in section 2, the joint probability density function (PDF) based on a Gaussian mixture can be used to derive the regression model for $K_{dp}$ estimation. The Gaussian mixture method (GMM) not only estimates the expected values of $K_{dp}$ by differentiating the conditional expectation of $\Phi_{dp}$, but also gives an estimation of $K_{dp}$ variance by regarding the errors in the calculation of the first derivative of $\Phi_{dp}$. In this section, we describe GMM for the $K_{dp}$ estimation using MZZU radar data. Figure 1 illustrates the flowchart of GMM (Fig. 1.b), comparing to that of the linear regression model (LR; Fig. 1.a).

From the chart of LR in Fig. 1.a, we can see that after the radar measurements are collected, the $\Psi_{dp}$ is unfolded, and then the clutter is removed. After these corrections, an iterative filtering method is applied to the $\Psi_{dp}$ profile. An adaptive method is finally used to estimate the $K_{dp}$ profile according to the values of $Z_H$. The Gaussian mixture model, on the other hand, processes $\Psi_{dp}$ differently. First of all, the clutter is masked out according to the thresholds of $Z_H$ and the variation of $\Psi_{dp}$. Secondly, the range $r$ and $\Psi_{dp}$ are fitted into a Gaussian mixture to yield the joint PDF, while the $\Psi_{dp}$ mean and the $\Psi_{dp}$ variance are obtained by taking the first raw and second central moments of the conditional PDF of $\Psi_{dp}$ given $r$. Thirdly, some specific clusters in the Gaussian mixture PDF are adjusted to solve the problems of ambiguous $\Psi_{dp}$ and backscattering

differential phase shift $\delta_{co}$ in order to derive the PDF of $\Phi_{dp}$. Fourthly, a raw $K_{dp}$ profile is calculated from the first derivative of the expected values of $\Phi_{dp}$, and the associated variances are obtained via a Taylor series expansion. Finally, the raw $K_{dp}$ profile is smoothed, and consequently, the variances are reduced. In addition, new $\Phi_{dp}$ with lower variances can be re-constructed from the $K_{dp}$ estimates.

## 4.1 Data masking

The presence of clutter in the $\Psi_{dp}$ measurements may severely affect the $K_{dp}$ estimation, producing significantly large variations on the estimates. It is well known that the effect of clutter can be reduced by applying a spectrum filter to the time-series data (e.g., May and Strauch, 1998; Hubbert et al., 2009). However, some residual clutter echoes are still shown on the radar measurements including $\Psi_{dp}$ (Wen et al., 2017). Therefore, the clutter needs to be well handled in GMM, prior to the deviation of the regression model based on the joint PDF.

In LR, the clutter is often eliminated by some criteria based on $\Psi_{dp}$ or $\rho_{hv}$. For instance, we use the thresholds of local standard deviation of $\Psi_{dp}$ less than $10°$ to classify valid points. Further, ten consecutive range gates of valid points signify the beginning of a rain cell, and five consecutive gates of invalid points finish the associated rain cell. Overall, the thresholds give a fairly good performance on the MZZU radar, however, the clutter may be incorrectly identified in the regions of high reflectivity or for the echoes mixed by weather and clutter, which are often associated with large $\Psi_{dp}$ variation.

In contrast, GMM adopts sophisticated procedures, as depicted in Figure 2. It is clear that there are five stages in the data masking, beginning with the input of raw $\Psi_{dp}$ and ending with masked data. At the first stage, the raw data are fitted to a Gaussian mixture initialized by the $k$-means clustering, while the covariance is set to be diagonal for simplicity. The clusters with no more than 5 points are promptly masked out, before they pass to the second stage. Stages two, three and four of the process all involve the clusters. At the second stage, the clusters are validated according to two sets of thresholds with respect to mean reflectivity. For the MZZU radar, the ratio of the standard deviations, $\sigma(\Psi_{dp})/\sigma(r)$, less than $14.2°$ km$^{-1}$, and $\sigma(\Psi_{dp})$ less than $4.1°$ are used for $Z_H$ less than 41 dBZ. To reduce the mis-classification in the hail regions, the thresholds are increased for higher $Z_H$, resulting in $\sigma(\Psi_{dp})/\sigma(r) < 47.9°$ km$^{-1}$ and $\sigma(\Psi_{dp}) < 6.3°$. Next, the entire $\Psi_{dp}$ profile is divided into multiple rain cell segments by considering the gaps between two consecutive clusters. Similar to the first stage, the segments containing no more than 5 points are excluded from the output of masked data. Following this, the dominant is determined for each segment by comparing the weight accumulations of weather and clutter clusters. For a clutter segment with mean height below 200 m, the clusters within the segment are re-evaluated by thresholds of $\sigma(\Psi_{dp})/\sigma(r) < 2.0°$ km$^{-1}$ and $\sigma(\Psi_{dp}) < 0.8°$, on the other hand, the clusters in a weather segment are re-examined using $\sigma(\Psi_{dp})/\sigma(r) < 34.7°$ km$^{-1}$ and $\sigma(\Psi_{dp}) < 6.1°$. This step can efficiently identify the clutter-contaminated weather echoes, which are often associated with large variances. At the last stage, some isolated points along the azimuth are obscured in the final results.

Figure 3 illustrates two examples for data masking, including a convective case (Fig. 3.a) and a stratiform case (Fig. 3.b). The data points in the two cases show steadily increasing trends related to anisotropic media along the wave propagation path. However, between 1.3 and 15 km at an azimuth of $252°$ in the convective case (Fig. 3.a), the data present significant fluctuations with the minimum value at about $0°$ but the maximum value at $180°$. Since the dynamic range of $\Psi_{dp}$ is from 0 to $180°$ for

the MZZU radar, the measurements near the ground are likely to be the clutter returns, verifying the results of data masking. After 15 km, the $\Psi_{dp}$ points start from about $50°$ and go all the way up to $180°$. Notwithstanding this trend, the points sharply decrease to about $10°$ at about 40 km, indicating the occurrence of phase folding. The data masking can effectively detect the phase folding, and provide valid masked data for deriving the joint PDF. On the other hand, the weather echoes are more

frequently observed at $1°$ in azimuth in the stratiform case (Fig. 3.b). By taking a closer inspection on the $\Psi_{dp}$ data, we can discern that the points largely fluctuate between 40 and 80 km due to low signal-to-noise ratio. In LR, these points may be incorrectly discarded based on $\sigma(\Psi_{dp})$ thresholds, leading to some missing data in the stratiform regions. In contrast, the data masking accurately identifies weather echoes characterized by a number of vertically-oriented density ellipses. The continuous and uniformly-distributed regimes are consistent with the physical interpretation of stratiform precipitation. In addition, the

data masking is also sensitive to sudden jumps at the beginning of the $\Psi_{dp}$ data, which may be caused by $\delta_{co}$.

## 4.2   $\Psi_{dp}$ density estimation

In the previous section, it is shown that the $\Psi_{dp}$ profile varies along the range $r$. It rises quickly for horizontally-oriented anisotropic scatterers, and conversely, it falls steadily for vertically-oriented particles (Marzano et al., 2010). The non-uniform beam filling is also an important source of errors for the $\Psi_{dp}$ measurements (Gosset, 2004; Ryzhkov, 2007).

To estimate the relationship between $r$ and $\Psi_{dp}$, we consider $r$ as an independent variable, denoted as $x$, and $\Psi_{dp}$ as a dependent variable, denoted as $y$. If the minimization of mean square error is required, the regression function is obtained by taking the average value of $y$ at fixed $x$, equivalent to estimating the expected values of $y$ conditioned on $x$, i.e.,

$$\bar{y}(x) = E(y|x) = \int y\, p(y|x, \beta)\, dx, \tag{10}$$

where $\beta$ is a set of unknown variables, for example, $\beta = (m, w, \mu, \Sigma)$ for the mixture model. Since the Gaussian mixture can

be used to model any shapes of probability density with a rapid speed, the $(x,y)$ points are then assumed to follow a joint PDF of Gaussian mixture, as defined in Eq. (4). Moreover, the properties of the multivariate Gaussian distribution in each cluster determine the Gaussianity of the marginal distribution of either variable and the conditional distribution of one variable given the other (Bishop, 2006). Therefore, the conditional PDF of $y$ given $x$ is expressed as

$$p(y|x, \beta) = \sum_{i=1}^{m} w_i^{y|x} \mathcal{N}\left(y; \mu_i^{y|x}, \Sigma_i^{y|x}\right), \text{ with} \tag{11}$$

$$\mu_i^{y|x} = \mu_i^y + \Sigma_i^{yx}(\Sigma_i^{xx})^{-1}(x - \mu_i^x), \tag{12}$$

$$\Sigma_i^{y|x} = \Sigma_i^{yy} - \Sigma_i^{yx}(\Sigma_i^{xx})^{-1}\Sigma_i^{xy}, \tag{13}$$

$$w_i^{y|x} = \frac{f_i(x)}{f(x)} = \frac{w_i \mathcal{N}(x; \mu_i^x, \Sigma_i^{xx})}{\sum_{j=1}^{m} w_j \mathcal{N}(x; \mu_j^x, \Sigma_j^{xx})}, \tag{14}$$

where $w_i$, $\mu_i = (\mu_i^x, \mu_i^y)^T$ and $\Sigma_i = \begin{pmatrix} \Sigma_i^{xx} & \Sigma_i^{xy} \\ \Sigma_i^{yx} & \Sigma_i^{yy} \end{pmatrix}$ are obtained by the EM algorithm. In Eq. (14), $f(x)$ is the marginal PDF of $x$ with the parameters identical to the mixture, and $f_i(x)$ is the weighted marginal PDF of each cluster, i.e., $f(x) =$

$\sum_{i=1}^{m} f_i(x)$. By substituting Eq. (11) into Eq. (10) and noting the linearity of the mathematical expectation, the expected value of $y$ conditioned on $x$ is then obtained as

$$E(y|x) = \sum_{i=1}^{m} \frac{f_i(x)}{f(x)}(a_i x + b_i), \text{ with} \tag{15}$$

$$a_i = \Sigma_i^{yx}(\Sigma_i^{xx})^{-1}, \tag{16}$$

$$b_i = \mu_i^y - \Sigma_i^{yx}(\Sigma_i^{xx})^{-1}\mu_i^x, \tag{17}$$

and the conditional variance is given as (see Appendix B)

$$\sigma^2(y|x) = \sum_{i=1}^{m} w_i^{y|x}\left[\Sigma_i^{y|x} + \left(\mu_i^{y|x}\right)^2\right] - \left(\sum_{i=1}^{m} w_i^{y|x}\mu_i^{y|x}\right)^2. \tag{18}$$

Equations (15) and (18) play an important role in the joint PDF-based regression analysis, called the regression and skedastic functions (Spanos, 1999). In Eq. (15), it can be seen that the regression function in GMM consists of multiple linear kernels, which is similar to LR. However, the weighting function $w_i^{y|x}$ is not determined by the local structure but the marginal PDF of global data $x$. The Gaussian mixture method is flexible to capture the data information, while it still retains a finite set of parameters. Moreover, Eq. (18) readily estimates the point-wise variances $\sigma^2(y|x)$ that characterize the random errors in the measurements. In contrast, Eq. (3) for the LR presents the relationship between the errors $\sigma(\Psi_{dp})$ and $\sigma(K_{dp})$ under the ideal conditions, which does not consider the random errors in $\Psi_{dp}$. It indicates that the GMM has a better error characterization based on the measurements when compared to the LR.

Figure 4 compares the $\Psi_{dp}$ profiles given by Eqs. (15) and (18) with that obtained by LR. Figure 4.a gives the same example as Fig. 3.a, but the EM algorithm is configured differently. In the $\Psi_{dp}$ density estimation, the mixture with full covariance yields density ellipses of random shapes. Furthermore, the algorithm repeats the fitting procedures three times to avoid the local maxima of the log-likelihood. Meanwhile, the choice of the cluster number relies on the Bayesian information criterion calculated for each $m$, starting at 10 clusters. It can be seen that the mixture composed by density ellipses well characterizes the data points, since the root-mean-square error is small relative to the expected values. Between 15 and 35 km, the narrow ellipses result in $\Psi_{dp}$ with a rising trend consistent with LR. On the other hand, the mixture has very small variances, giving a high confidence for the fitted parameters. From 35 km, the ellipses become wider, and the associated variances increase due to low signal-to-noise ratio at the edge of radar echoes. What is notable, however, is that the $\Psi_{dp}$ profile dramatically increases to a large value, whereas LR remains a relatively steady trend. It indicates the importance of the $\Psi_{dp}$ unfolding for the $\Psi_{dp}$ density estimation.

Figure 4.b presents another example of the density estimation. It is clear that the $\Psi_{dp}$ profiles produced by GMM and LR both rise considerably along the range, and the trends for the two methods are very similar with a strong correlation of 0.998. The profile starts at about $50°$, and remains relatively stable before rising dramatically between 35 and 55 km. By 65 km, $\Psi_{dp}$ has more than doubled, and then, there is a steady increase for $\Psi_{dp}$ reaching about $130°$ at the end of the profile, which is around $70°$ up on the ranges of 0 and 35 km, and $10°$ more than recorded at the ranges of 55 and 65 km. If we examine $\Psi_{dp}$ measured at X-band frequency, we can see that some points fall out of the dash lines corresponding to one standard error (i.e.,

95% interval). Most notably, the $\Psi_{dp}$ profile shows a sudden slump between 18 and 20 km, while the $Z_{DR}$ corresponds to a local peak (not shown). It may indicate the occurrence of $\delta_{co}$. In conclusion, the expected value and the variance of $\Psi_{dp}$ can be obtained from the joint PDF, but the mixture needs to be tuned in terms of $\Psi_{dp}$ unfolding and $\delta_{co}$ elimination in order to obtain the PDF of $\Phi_{dp}$.

### 4.3  $\Psi_{dp}$ unfolding and $\delta_{co}$ elimination

According to the continuity and consistency of the phase data, we can discern that some issues exist in the density estimation, such as ambiguous $\Psi_{dp}$ and $\delta_{co}$. Since $\Psi_{dp}$ is an range accumulative measurement of propagation phase, depending on the initial $\Psi_{dp}(0)$. The measurements may exceed the dynamic range of 0–180° when the wave propagates through a rain medium. This situation is even more significant at X-band frequency than S-band due to the inverse relation of the wavelength and the
rate of phase shift. Nevertheless, it can be noted that $\Psi_{dp}$ gives a non-negative trend along the range for rain, and therefore, the ambiguous $\Psi_{dp}$ may be corrected accordingly (Wang and Chandrasekar, 2009).

In LR, $\Psi_{dp}$ is first averaged over a small window for weather data, and a linear fit is then performed to obtain the increment for the range gate next to the window. In the following stage, a reference is predicted by summing up the average and the increment, and compared to the observed value at the same gate. If the difference between the predicted and observed values
is larger than 90°, the observed $\Psi_{dp}$ is then increased by 180°. Finally, the correction process is iteratively operated until the last gate.

On the other hand, the $\Psi_{dp}$ unfolding is more straightforward in GMM. Figure 5 shows the flowchart of the $\Psi_{dp}$ unfolding and the $\delta_{co}$ elimination. After obtaining the PDF of $\Psi_{dp}$, the initial step of the $\Psi_{dp}$ unfolding selects the density ellipses with at least 6 data points. Next, the second step calculates the difference of the means $\mu_i$ between the two consecutive density ellipses
along the range. At this point, the PDF of $\Psi_{dp}$ is ready to be corrected for ambiguous $\Psi_{dp}$. In the final step, the mean of the latter density ellipse is added up 180°, if the former mean is larger than the latter one by 80°.

As illustrated in Fig. 4.a, the profile $\Psi_{dp}$ reaches 180° at about 38 km, and then becomes ambiguous between 38 and 42 km. In LR, the $\Psi_{dp}$ values at these locations are interpolated according to the trend of the previous few gates, and the maximum value is 180°. In contrast, the corrected density ellipses in GMM show an upward trend between 38 and 42 km, while the $\Psi_{dp}$
profile reaches a maximum value of about 195°, indicating the effectiveness of the $\Psi_{dp}$ unfolding in the region of heavy rain. In addition, when we apply the algorithm to a larger dataset, the rate of false alarm reaches a very small value at 0.66%.

In addition to ambiguous $\Psi_{dp}$, the estimation of the joint PDF may also be affected by non-zero $\delta_{co}$, which is defined as the phase difference between the horizontal and vertical polarizations upon the backscattering of the particles in a radar resolution volume. This effect occurs more frequently at X-band frequency than S-band due to Mie scattering (Trömel et al., 2013). The
$\delta_{co}$ is shown as a sudden phase change over a small number of gates in a monotonically increasing trend for rain. According to this manifestation, the magnitude and gate number of the $\Psi_{dp}$ perturbation can be used to eliminate $\delta_{co}$ (Matrosov et al., 2002; Otto and Russchenberg, 2011).

The linear regression model often adopts an iterative filter technique, which generates a new $\Phi_{dp}$ profile from either the raw data or the filtered one based on a threshold (Hubbert and Bringi, 1995). If the filtering alters the data by 4°, the new profile

selects the filtered data, otherwise the raw data are remained. The new profile is then used as input in the next iteration until the convergence condition is satisfied.

As shown in Fig. 5, the $\delta_{co}$ elimination is embedded into the process of the $\Psi_{dp}$ unfolding. For two consecutive density ellipses, the latter density ellipse is removed if its mean is larger than the former one by 85°. Prior to this step, the mean of the first density ellipse in the mixture should be below 90° to reduce the $\delta_{co}$ effect at the first few gates. Since $\delta_{co}$ occurs over a small number of range gates, a mixture pruning is also employed to remove the density ellipses with weights less than 0.0501, equivalent to 2% of the data.

It is clear from Fig. 4.b that $\delta_{co}$ has occurred at multiple locations in the data. The $\Psi_{dp}$ profile starts at a high value and drops somewhat over the first two gates. Notably, there is a narrow gap between 18 and 20 km, while the corresponding $Z_{DR}$ presents a local peak. It signifies that non-zero $\delta_{co}$ occurs in this region. In GMM, these data are characterized by a density ellipse with a slightly decreasing trend, and the resulting expected values are consistent with the filtered data in LR. Between 70 and 90 km, a few isolated points beyond the density ellipses are associated with $\delta_{co}$. Both of the two methods can produce $\Phi_{dp}$ following the main trend of the data, which suggests that the process is effective for the $\delta_{co}$ elimination.

## 4.4 $K_{dp}$ density estimation

As discussed in section 2.1, $K_{dp}$ is the first derivative of $\Phi_{dp}$ with respect to the range $r$. According to the mean value and dominated convergence theorems, the derivative of the expected value of $\Phi_{dp}$ conditioned on $r$ is equal to the expected value of the derivative of $\Phi_{dp}$ with respect to $r$, i.e., $K_{dp}$ (see Appendix C). Following the notation in section 4.2, we denote $K_{dp}$ as $y'$. Therefore, the expected value of $K_{dp}$ is obtained by taking the derivative of Eq. (15), yielding

$$E\left(y'|x\right) = \frac{1}{f^2(x)}\left\{\sum_{i=1}^{m}\sum_{j=1}^{m}f_i(x)f_j(x)\left[\left(\frac{x-\mu_j^x}{\Sigma_j^{xx}}-\frac{x-\mu_i^x}{\Sigma_i^{xx}}\right)(a_ix+b_i)+a_i\right]\right\}. \tag{19}$$

The variance of $y'$ conditioned on $x$ can be approximated by the first-order Taylor series expansion (see Appendix D), i.e.,

$$\sigma^2\left(y'|x\right) = \left[E''(y|x)\right]^2\sigma^2(y|x), \tag{20}$$

where $\sigma^2(y|x)$ is given in Eq. (18). By taking the derivative of Eq. (19), $E''(y|x)$ is expressed as

$$E''(y|x) = 2\left[\sum_{i=1}^{m}a_i\left(w_i^{y|x}\right)'\right]+\sum_{i=1}^{m}(a_ix+b_i)\left(w_i^{y|x}\right)''. \tag{21}$$

From Eq. (C8) in Appendix C, it is clear that

$$\left(w_i^{y|x}\right)' = \frac{g_i(x)}{f^2(x)} = \frac{1}{f^2(x)}\sum_{j=1}^{m}f_j(x)f_i(x)\left(\frac{x-\mu_j^x}{\Sigma_j^{xx}}-\frac{x-\mu_i^x}{\Sigma_i^{xx}}\right), \tag{22}$$

where $g_i(x)$ is the summation term. Subsequently, the second derivative of $w_i^{y|x}$ is given as

$$\left(w_i^{y|x}\right)'' = \frac{g_i'(x)f(x) - 2f'(x)g_i(x)}{f^3(x)}, \quad \text{where} \tag{23}$$

$$f'(x) = -\sum_{j=1}^{m}\left(\frac{x-\mu_j^x}{\Sigma_j^{xx}}\right)f_j(x), \tag{24}$$

$$g_i'(x) = \sum_{j=1}^{m} f_j(x)f_i(x)\left[\left(\frac{x-\mu_i^x}{\Sigma_i^{xx}}\right)^2 - \left(\frac{x-\mu_j^x}{\Sigma_j^{xx}}\right)^2 + \frac{1}{\Sigma_j^{xx}} - \frac{1}{\Sigma_i^{xx}}\right]. \tag{25}$$

Equations (19) and (20) are the regression and skedastic functions for the $K_{dp}$ estimation. In Eq. (19), it is clear that the expected value of $K_{dp}$ can be divided into two components, including Eqs. (C7) and (C11). On one hand, Eq. (C7) is related to the changing rate $a_i$ weighted by the marginal distribution of each cluster in the mixture, equivalent to a linearly weighted combination of small portions of data. If a data point is dominated by a specific cluster, i.e., the weight of a cluster is significantly larger than the others, $K_{dp}$ is determined by the coefficients of the cross-correlation and auto-correlation of $r$, and independent of the means and auto-correlation of $\Phi_{dp}$, yielding a constant value within the dominated cluster. On the other hand, Eq. (C11) shows that the weighting function also contributes to the $K_{dp}$ estimates by considering the Gaussian derivative of the $\Phi_{dp}$ estimates in two or three adjacent clusters along the range. The sign of $K_{dp}$ is then determined by the marginal means and variances of the clusters, weighted by the difference of their contributions to $\Phi_{dp}$.

In Eq. (20), it can be seen that $\sigma^2(K_{dp})$ is proportional to $\sigma^2(\Phi_{dp})$, which is similar to Eq. (3). When the LR is applied to the MZZU radar, we often assume that $\sigma(\Phi_{dp})$ is equal to $2.61°$ with 32 pulses, 1 m s$^{-1}$ for Doppler spectrum and 0.98 for $\rho_{hv}$ under the ideal conditions. However, in the GMM, $\sigma^2(\Phi_{dp})$ varies along the range due to the random errors of the $\Phi_{dp}$ estimates, indicating the GMM can provide a better model for the error characteristics in the $\Phi_{dp}$ measurements. In addition, the statistical errors with respect to signal processing may be included in Eq. (20) as an additive term, independent of $\Phi_{dp}$. Moreover, $\sigma^2(K_{dp})$ in GMM is closely related to the first derivative of $K_{dp}$ in Eq. (20). As the changing rate of $K_{dp}$ increases, the random errors associated with the $K_{dp}$ estimates rise dramatically.

Figure 6.b illustrates $K_{dp}$ and its variance estimated from $\Phi_{dp}$ in Fig. 6.a, which is the same case as given in Figs. 3.a and 4.a. It is apparent that the $K_{dp}$ estimates present a large fluctuation, while the associated variances are significant. In GMM, $K_{dp}$ starts from about 0.5 deg km$^{-1}$, and then fluctuates between 17 and 20 km and between 24 and 42 km. In the profile, there are six local peaks with the maximum at about 8.5 deg km$^{-1}$. Meanwhile, the $K_{dp}$ variances vary as the $K_{dp}$ estimates change. Between 15 and 17 km and between 20 and 24 km, the $K_{dp}$ estimates stand at a value, leading to small $K_{dp}$ variances in these regions. When short excursions are present, such as that between 18 and 20 km, $K_{dp}$ variances increase significantly due to the contribution of the first derivative of $K_{dp}$ in Eq. (20). Furthermore, the large $\Phi_{dp}$ variances between 35 and 42 km also result in an increase of the $K_{dp}$ variances. In contrast, LR gives less fluctuation in $K_{dp}$ estimates with two peaks at about 20 and 34 km. The comparison of $K_{dp}$ obtained by the two methods may suggest that a smoothing procedure is required to reduce the significant variance in GMM.

## 4.5 $K_{dp}$ smoothing

As discussed previously, the $K_{dp}$ variance is small for high $K_{dp}$, but relatively large for low $K_{dp}$. Therefore, an adaptive estimation is adopted in LR. For radar reflectivity ($Z_H$) less than 20 dBZ, the gate number $n$ in Eq. (2) is set as 15, while $n$ is 8 for $20 \leq Z_H < 35$ dBZ, and 2 for $Z_H \geq 35$ dBZ, respectively. On the other hand, GMM also applies an adaptive technique

based on finite impulse filter (FIR) to the expected values of $K_{dp}$ in order to reduce the associated variances. Figure 7 shows the time responses of the FIR with the cutoff frequency of 0.053 and the Gaussian window of 28, which yield the best performance for the MZZU radar. The impulse response (Fig. 7.a) is peaked at the center, and gradually decreases towards the two ends. Furthermore, the step response (Fig. 7.b) gives the accumulation of the impulse response, indicating that the magnitudes around the center change faster than that at the two ends. If a longer window is required, the order of the FIR is increased accordingly.

In this study, we gradually increase the order number to calculate the difference between the $K_{dp}$ profiles obtained by the FIR filters with two adjacent order numbers. The optimal order of the FIR filter is then set when the relative square error of the two $K_{dp}$ is below 0.001. For profiles with sufficiently large data points, the order number is between 29 and 33 for the MZZU radar.

To obtain the reduced variance, we consider the filter as a number of weighting functions, denoted as $h_i(x)$, and subsequently,

the smoothed data become

$$y = \sum_{i=1}^{n} h_i * x_i \tag{26}$$

where $y$ is a smoothed data point, $x_i$ is the original data within the smoothing window, and $n$ is the window length. By taking the variance on both sides of Eq. (26), we have

$$\sigma^2(y) = \sum_{i=1}^{n} h_i^2 \sigma^2(x_i). \tag{27}$$

Therefore, the variance of the smoothed data is the weighted sum of the variances of the original data within the smoothing window. Since the FIR coefficients are much less than unity, $\sigma^2(y)$ is smaller than $\sigma^2(x)$ at the same gate. Furthermore, the $K_{dp}$ estimates with the reduced variances can be used to re-construct $\Phi_{dp}$ to obtain smaller $\Phi_{dp}$ variances. For a fixed gate spacing $\Delta r$, the re-constructed $\Phi_{dp}$ for the $j$th range gate is

$$\Phi_{dp}^j = \sum_{i=1}^{j} K_{dp}^i \Delta r, \quad \text{and} \tag{28}$$

$$\sigma^2(\Phi_{dp}^j) = \sum_{i=1}^{j} \sigma^2(K_{dp}^i) \Delta r^2. \tag{29}$$

The red curves in Figs. 6.a and 6.b illustrate the re-constructed $\Phi_{dp}$ and the smoothed $K_{dp}$ using FIR, respectively. The smoothed $K_{dp}$ in Fig. 6.b is more consistent with the LR results compared to the original $K_{dp}$ produced by the GMM. In the first few kilometers, the smoothed $K_{dp}$ gradually rises, and then peaks at about 21 km. With no fluctuations, the smoothed $K_{dp}$ falls gradually, followed by a growth before reaching a plateau at 33 km. After a slight decrease between 33 and 36 km, $K_{dp}$

rises dramatically, which is very different from LR. Meanwhile, the variances are small at the beginning, but get larger as $K_{dp}$ is climbing up. Between 20 and 33 km, the $K_{dp}$ estimates do not change very much, leading to small variances in this region. But after 33 km, the variances begin to increase and retain large values until the end of the profile. Overall, the smoothed $K_{dp}$ is stable, producing a profile considerably consistent with LR, and the variances are significantly reduced comparing

to the original data. In addition, the re-constructed $\Phi_{dp}$ (Fig. 6.a) constantly increases with few local fluctuations, while the associated variances are smaller than the $\Phi_{dp}$ variances in GMM.

## 5   Evaluation

In this section, a case study is first presented to qualitatively analyze the storm structure and evolution based on $K_{dp}$. The radar-gauge dataset is then used to provide a quantitative evaluation for the $K_{dp}$ estimation in terms of root mean squared error

(RMSE), normalized bias (NB) and Pearson correlation coefficient ($\rho_{RG}$), which are defined as

$$\text{RMSE} = \sqrt{\frac{\sum_{i=1}^{N}(R_i - G_i)^2}{N}}, \tag{30}$$

$$\text{NB} = \frac{\sum_{i=1}^{N}(R_i - G_i)}{\sum_{i=1}^{N} G_i}, \tag{31}$$

$$\rho_{RG} = \frac{\sum_{i=1}^{N}(R_i - \bar{R})(G_i - \bar{G})}{\sqrt{\sum_{i=1}^{N}(R_i - \bar{R})^2}\sqrt{\sum_{i=1}^{N}(G_i - \bar{G})^2}}, \tag{32}$$

where $N$ is the sample size, $R_i$ is the individual radar hourly rain amount, $G_i$ is the gauge data, and $\bar{R}$ and $\bar{G}$ are the sample

means for radar and gauge, respectively. The radar hourly rain amount is calculated based on the CASA radar rainfall algorithm. It is given as (Wang and Chandrasekar, 2010; Chen and Chandrasekar, 2015)

$$R(K_{dp}) = 18.15 K_{dp}^{0.79}, \tag{33}$$

where $R$ is the instantaneous rain rate in mm h$^{-1}$. It is noted that the radar collects instantaneous measurements every 4–5 minutes, whereas RGs obtain the precipitation accumulations over 60 minutes. Therefore, it is necessary to average 12–15

consecutive radar scans to derive the hourly rain amounts.

### 5.1   Case study

On 24 March 2016, a severe storm developed in central Missouri and moved eastward across Columbia, MO, causing strong winds and heavy precipitation at the surface. When the storm became mature, the S-band radars at Kansas City and St. Louis observed the storm structure at high levels, since each radar was about 150 km away from the storm. Notably, the Kansas City

radar showed positive and negative Doppler velocities in a small area (not shown), indicating the occurrence of a downburst. On the other hand, the MZZU radar illustrated a bow echo of $Z_H$ close to the radar center (Figs. 8.b). In addition to $Z_H$, the GMM-based $K_{dp}$ (Figs. 8.d, e and f) was also obtained to investigate the storm structure near the surface.

Figure 8 illustrates that the convective storm evolves from a strong and large echo to a bow shape echo, and then dissipates at far range. At 0304 UTC (Fig. 8.a), a cell with strong $Z_H$ moves into the radar area, while $K_{dp}$ is moderate with a maximum of about 3 deg km$^{-1}$ (Fig. 8.d). As the cell is transforming to a bow shape, the radar echo becomes intensive, and forms a rain band with embedded convective cores (Fig. 8.b). It is clear to see that $K_{dp}$ reaches over 10 deg km$^{-1}$ in these core regions (Fig. 8.e), indicating very heavy precipitation at the surface. With the fast movement of the storm, the downburst has been weaken, and the storm starts to dissipate (Fig. 8.c). At 0441 UTC, it can be seen that $K_{dp}$ is gradually reduced at the far range, while its maximum is much less than that at the mature stage.

In this storm, the bow echo is shown as a number of convective cores embedded in a rain band, while the downbursts occured at the leading edge near the echo center. The bow echo can be considered as a mesoscale convection with a horizontal dimension of more than 60 km. To gain a further insight, Fig. 9 shows raw $\Phi_{dp}$ and $K_{dp}$ for the bow echo. In Fig. 9.a, raw $\Phi_{dp}$ presents large gradients along the leading edge, rising from about 50° to over 140°. Due to the sharp increase, $\Phi_{dp}$ exceeds the maximum dynamic range, leading to ambiguity in the areas of X:-20~-18 km and Y:12~18 km and X:-40~-23 km and Y:-8~-5 km. In addition, the echoes behind the convective cores occasionally vanish as a result of signal attenuation. Nevertheless, LR (Fig. 9.b) produces continuous $K_{dp}$ by $\Phi_{dp}$ unfolding and linear interpolation according to the trends of the profiles, but some missing data still exist within the storm, due to low signal-to-noise ratio. In contrast, GMM (Fig. 9.c) corrects these data with the expected values derived from the joint PDF, and simultaneously obtains the statistical errors in the production of $K_{dp}$. It is evident that the GMM method can efficiently handle the missing data via the mixture model, which is another advantage over the LR model. Furthermore, the statistical errors are not very large in these areas, since the missing data are filled by the distribution of the entire data profile. Additionally, the GMM $K_{dp}$ estimates are generally a few deg km$^{-1}$ higher than the LR ones, particularly for the regions of high $Z_H$.

By taking a closer look at GMM $K_{dp}$, we can see that the bow echo is generally characterized by $K_{dp}$ of above 2.5 deg km$^{-1}$, while five pockets of high $K_{dp}$ are identified. In the bow head, the first pocket presents very high $K_{dp}$ associated with a rapid growth of $\Phi_{dp}$. Behind this pocket, there is a region of negative $K_{dp}$, whereas LR generally yields positive values. It may be due to a reduction of cross-correlation coefficient caused by low signal-to-noise ratio, since the signals have been significantly attenuated after propagating through the pocket. In the middle of the second and third pockets in the bow center, LR and GMM both show lower $K_{dp}$ comparing to the two pockets, while $K_{dp}$ is substantially consistent with the gradient of $\Phi_{dp}$ in the area. By considering the high $Z_H$ in Fig. 8, these moderate $K_{dp}$ values may indicate less anisotropic scatterers, such as small hail in the process of wet growth. Similarly, a hail signature with maximum $Z_H$ of above 66 dBZ and small $K_{dp}$ of 1~2 deg km$^{-1}$ can also be identified in the middle of the fourth and fifth pockets in the bow tail. Along with the expected values of GMM $K_{dp}$, Fig. 9.d depicts the statistical errors $\sigma(K_{dp})$ in the calculation of the expected values. The five pockets of high $K_{dp}$ are generally associated with small $\sigma(K_{dp})$ of a few tenths deg km$^{-1}$. However, the estimates behind the top four pockets yield very large $\sigma(K_{dp})$ with a maximum above 10 deg km$^{-1}$, and the expected values of $K_{dp}$ are sometimes below 0 deg km$^{-1}$, such as X:-25~-20 km and Y:11~20 km. In contrast, a region of high $\sigma(K_{dp})$ appears in front of the bottom pocket, superimposed on the high $Z_H$ area associated with hail. In conclusion, the GMM $K_{dp}$ estimates of high confidence give good

agreement with the gradients of $\Phi_{dp}$ in the leading edge of the bow echo, while large $\sigma(K_{dp})$ are expected at the region of high variation of the $K_{dp}$ estimates.

To give a further evaluation of the GMM $K_{dp}$, Fig. 10 compares the scatterplots of $Z_H$, $Z_{DR}$ and $K_{dp}$ to the self-consistency (SC) relations. Referring to Park et al. (2005b), Otto and Russchenberg (2011) and Matrosov (2010), the X-band SC relations are given as

$$Z_{DR} = \begin{cases} 0 & Z_H \leq 9.5 \\ 0.051Z_H - 0.486 & 9.5 < Z_H \leq 55 \\ 2.319 & Z_H \geq 55 \end{cases}, \tag{34}$$

$$K_{dp} = 1.37 \times 10^{-3} \times 10^{0.068Z_H} \times 10^{-0.042Z_{DR}}, \tag{35}$$

$$K_{dp}/Z_h = 1.2 \times 10^{-4} - 4.1 \times 10^{-5} Z_{DR} \quad (Z_{DR} < 1.6), \tag{36}$$

where $Z_H = 10 \log Z_h$ and $Z_h$ is in mm$^6$m$^{-3}$. Figures 10.a and b illustrate that the points concentrate at the region with $Z_H$ between 10 and 40 dBZ, where the $K_{dp}$ shows a low and steady increase. Both the LR $K_{dp}$ and GMM $K_{dp}$ agree well with the SC relation in Eqs. (34) and (35). It is notable that the $K_{dp}$ rises dramatically from a few tenths to 8 deg km$^{-1}$ for $Z_H$ larger than 40 dBZ. As depicted in Fig. 10.a, the LR $K_{dp}$ increases greatly when $Z_H$ reaches 50 dBZ, showing a difference from the SC relation. In contrast, the GMM $K_{dp}$ in Fig. 10.b gives some improvements over the LR $K_{dp}$ in Fig. 10.a when compared to the SC relation. Furthermore, the points of $Z_{DR}$–$K_{dp}/Z_h$ in Figs. 10.c and d may be grouped into two clusters with high populations. The cluster with lower $K_{dp}/Z_h$ agrees with the SC relation in Eq. (36). On the other hand, the cluster centered at $Z_{DR}$ around 0 dB are likely caused by hails, since they are less anisotropic than raindrops with the same size. In addition, the LR $K_{dp}$ and GMM $K_{dp}$ produce a similar distribution of $Z_{DR}$ and $K_{dp}/Z_h$, though the distribution for the GMM $K_{dp}$ tends to be narrower.

Moreover, the computational time is crucial for the real-time application of the $K_{dp}$ retrieval algorithms. For processing the data in Fig. 9 in Window 10 on a PC or Linux 7 on a supercomputer, the GMM takes about 7.058/4.068 seconds to process the $K_{dp}$ with/without the data masking, whereas the LR reduces the time to about 2.037 seconds. It indicates that the LR has the advantages of simplicity and efficiency. Nevertheless, the GMM can obtain more information from the radar data, which is useful for the model studies.

## 5.2 Statistical analysis

In order to quantitatively evaluate the accuracy of GMM $K_{dp}$, hourly accumulated rain amounts are derived from the X-band rainfall rate algorithm (Chen and Chandrasekar, 2015), and compared to the rain gauge data collected at Bradford, Sanborn, Auxvasse and Williamsburg between 1 April 2016 and 2 June 2018. The scatterplots presented in Fig. 11 illustrate

the comparison between GMM-based radar and gauge rain amounts, and the accompanying table (Table 2) gives RMSE, NB and $\rho_{RG}$ results obtained by GMM and LR.

Consistent with the data in Table 1, the rainfall at the four sites is predominately made up of light rain with hourly rain amounts no more than 2.5 mm h$^{-1}$. Nevertheless, according to Fig. 11, moderate rain with amounts between 2.6 and 8 mm h$^{-1}$ gives a considerable contribution to the total rain events, followed by a small portion of heavy rain with amounts more than 8 mm h$^{-1}$. When we study the scatterplots and statistics for each of the four sites, it is apparent that Bradford (Fig. 11.a and b) and Sanborn (Fig. 11.c and d) are more concentrated on the red line than Auxvasse (Fig. 11.e and f) and Williamsburg (Fig. 11.g and h), since Bradford and Sanborn are closer to the radar. Accordingly, RMSEs for Bradford and Sanborn (Table 2) are relatively small, about 13%~35% lower than Auxvasse and Williamsburg. Furthermore, it can be seen that Bradford and Sanborn show negative bias associated with negative NBs, indicating an underestimation of rain amounts by GMM $K_{dp}$. In contrast, a slight overestimation may be concluded for Auxvasse and Williamsburg by considering the point trends and the positive NBs. Additionally, Sanborn claims the highest $\rho_{RG}$ out of the four sites, yielding the best consistency between radar and gauge.

When compared to LR statistics as given in Table 2, it is clear that GMM improves the RMSEs, NBs and $\rho_{RG}$ for Auxvasse and Williamsburg. Notably, the GMM-based NB for Auxvasse reaches a very small value of 0.04, one fifths of LR-based NB. For Bradford, RMSE is reduced by GMM, but the absolute value of NB is slightly increased, while $\rho_{RG}$ remains the same. On the other hand, for Sanborn, the GMM-based RMSE, NB and $\rho_{RG}$ get worse by a few hundredths of millimeters, which may be due to the local complex terrain near the radar. Overall, the rainfall estimates of GMM $K_{dp}$ give a better performance than that of LR in terms of RMSE, NB and $\rho_{RG}$ at the far ranges.

To improve the accuracy of the radar rainfall estimation, we have optimized the $R$-$K_{dp}$ relation in terms of RMSE using the radar-gauge dataset. It leads to the relation as

$$R(K_{dp}) = 17.33 K_{dp}^{0.92}. \tag{37}$$

Figure 12 shows the scatterplots of the radar-gauge data and the statistics of RMSE, NB and $\rho_{RG}$ obtained by Eq. (37) for all the four sites. It is clear that Eq. (37) has improved the negative trend in Eq. (33). As illustrated in Figs. 12.a and b, the points give a better concentration on the one-to-one reference line with noticeable changes for higher $R(K_{dp})$. In Fig. 12.a, the LR $R(K_{dp})$ achieves fairly good RMSE, NB and $\rho_{RG}$ at 2.30, 0.02 and 0.80, respectively. On the other hand, the GMM $R(K_{dp})$ presents a similar distribution as the LR $R(K_{dp})$, but the points in Fig. 12.b are shifted toward the vertical axis. Moreover, the GMM $R(K_{dp})$ gives better RMSE and $\rho_{RG}$ at 2.22 and 0.81, respectively, and slightly worse NB at -0.03.

It can be found that the rain rates based on the GMM $K_{dp}$ have a moderate consistency with the rain gauge data. To further improve the results, some advanced rain rate algorithms can be considered, such as the rain-ice separation technique in the IFloodS campaign (Chen et al., 2017b) and the radar-gauge comparison method in the MC3E campaign (Giangrande et al., 2014). Nevertheless, the GMM has the advantage over the existing methods, since it can yield the variance of $K_{dp}$. Furthermore, the variance of $R$ can also be obtained by the $K_{dp}$ mean and the $K_{dp}$ variance via the $R$–$K_{dp}$ relation, leading to the variability in the error characteristics of $R$. Thus, the variances can be used to study the streamflow trends in the hydrological model.

## 6 Summary and discussions

In this study, we proposed a probabilistic method based on Gaussian mixture model to estimate the specific differential phase $K_{dp}$, which is the range derivative of differential phase shift $\Phi_{dp}$. The Gaussian mixture method (GMM) not only obtained the expected values of $K_{dp}$ by differentiating the conditional expectation of $\Phi_{dp}$, but also yielded the variance $\sigma^2(K_{dp})$ regarding the errors in the calculation of the first derivative of $\Phi_{dp}$.

As an initial step of GMM, the data masking was performed to eliminate the residual clutter in the measurements of the total differential phase ($\Psi_{dp}$). The data of $r$ and $\Psi_{dp}$ were first fitted into a simplified Gaussian mixture to generate a number of clusters, which were validated against the two sets of the $\sigma(\Psi_{dp})$ and $\sigma(\Psi_{dp})/\sigma(r)$ thresholds given by radar reflectivity $Z_H$. The clusters were then combined to form the rain cell segments, and the segments were classified by comparing the weight accumulations of weather and clutter clusters. Next, the clusters within each segment were re-evaluated by the thresholds according to the segment types. Finally, the azimuthally isolated points were masked out.

Secondly, the joint probability density function (PDF) was obtained by fitting the data of $r$ and $\Psi_{dp}$ into a mixture model with full covariance, where the cluster number $m$, weight $w$, mean $\mu$ and covariance $\Sigma$ were optimized via the Expectation-Maximization (EM) algorithm. Subsequently, the PDF of $\Psi_{dp}$ conditioned on $r$ was also a mixture with parameters related to the joint PDF. Finally, the $\Psi_{dp}$ mean was estimated by the conditional expectation, and the statistical errors $\sigma^2(\Psi_{dp})$ were given by the conditional variance, which was not always constant, but varied with $w$ and the marginal PDF of $r$.

Thirdly, the ambiguous $\Psi_{dp}$ and backscattering differential phase shift $\delta_{co}$ were corrected by examining the two adjacent density ellipses in the mixture. On one hand, if the former density ellipse had a mean larger than the latter one by $80°$, the latter mean was added to $180°$ for $\Psi_{dp}$ unfolding. On the other hand, if the former mean was smaller than the latter one by $85°$, the latter density ellipse was removed as $\delta_{co}$. Moreover, for $\delta_{co}$ elimination, the first density ellipse mean was assumed as below $90°$, while the density ellipses with small weights were also removed.

Fourthly, the joint PDF of $r$ and $\Phi_{dp}$ was used in the calculations of $K_{dp}$ and $\sigma^2(K_{dp})$. Since $K_{dp}$ was the range derivative of $\Phi_{dp}$, the expected values of $K_{dp}$ were then obtained via the derivative of the expected value of $\Phi_{dp}$. Moreover, by taking the first-order Taylor series expansion, $\sigma^2(K_{dp})$ was the product of the square of the first derivative of $K_{dp}$ and $\sigma^2(\Phi_{dp})$, yielding non-constant values of $\sigma^2(K_{dp})$.

In the final step, the expected values of $K_{dp}$ were smoothed to reduce the associated $\sigma^2(K_{dp})$. An FIR filter was implemented, and iteratively applied to the data to search for an optimal window length. Subsequently, the reduced $\sigma^2(K_{dp})$ was obtained by the sum of the original $\sigma^2(K_{dp})$ weighted by the FIR coefficient squares within the window. Additionally, new $\Phi_{dp}$ were re-constructed from the smoothed $K_{dp}$, while $\sigma^2(\Phi_{dp})$ was also reduced.

The experimental results with a severe storm observed by the X-band polarimetric radar in the University of Missouri (MZZU) revealed the advantages of GMM. By studying the structure and evolution of a bow echo in the storm, it was concluded that the GMM $K_{dp}$ was consistent with the gradients of raw $\Phi_{dp}$ along the leading edge of the bow echo, while large $\sigma^2(K_{dp})$ occurred with high variation of $K_{dp}$. The GMM method produced results similar to the LR method, with the ability to handle the missing data. Moreover, the hourly rain amounts based on $K_{dp}$ were compared to the rain gauge data, showing fairly good

agreement between radar and gauge measurements. The rain amounts obtained by GMM $K_{dp}$ gave improvements over the linear regression model, particularly for the far ranges.

The potential applications of GMM $K_{dp}$ and $\sigma^2(K_{dp})$ include quantitative precipitation estimation (Cifelli et al., 2011; Chen et al., 2017a) and attenuation correction (Park et al., 2005a). For quantitative precipitation estimation, the relationship between $K_{dp}$ and rain rate $R$ is almost linear, since $K_{dp}$ is about the fourth-order moment of raindrop size distribution, and $R$ is the 3.67th-order moment. As illustrated in Figs. 11 and 12, the $R(K_{dp})$ algorithm is consistent with the in situ measurements. To further investigate the $R$ errors, it is necessary to consider the $K_{dp}$ errors in the calculation of the first derivative of $\Phi_{dp}$. The standard deviation $\sigma(K_{dp})$ is then related to $\sigma(R)$ by a factor of $R/K_{dp}$ (Bringi and Chandrasekar, 2001). In a similar manner, $K_{dp}$ is linearly proportional to specific attenuation $A_H$ and specific differential attenuation $A_{DP}$ (Bringi and Hendry, 1990). Therefore, the errors of radar reflectivity $Z_H$ and differential reflectivity $Z_{DR}$ may also be proportional to $\sigma(K_{dp})$ after the attenuation correction, and eventually contribute to the $R$ errors via $R(Z_H)$ and $R(Z_H, Z_{DR})$. Moreover, the error estimates can be used to study the propagation of uncertainty in the weather model and provide streamflow trends in hydrological model. In the future study, the algorithm will also be extended to other frequencies, such as C-band (Vulpiani et al., 2012; May et al., 1999b) and S-band (Bringi and Chandrasekar, 2001). The thresholds in the data masking, the $\Psi_{dp}$ unfolding and the $\delta_{co}$ elimination will be adjusted according to the radar specifications. Nevertheless, the steps for the calculations of the PDFs of $\Psi_{dp}$ and $K_{dp}$ will be remained.

## Appendix A: Regression-based estimation of $K_{dp}$

Let the total differential phase $\Psi_{dp}$ be $y$, and the range gate $r$ be $x$. The $\Psi_{dp}$ profile over small range segments can be approximated by a first-order polynomial, i.e,

$$y = \beta_0 + \beta_1 x + \epsilon, \tag{A1}$$

where $\beta_0$ and $\beta_1$ are the coefficients in the linear approximation, and $\epsilon$ is an error function. It can be assumed that $\epsilon$ is independent and individual distributed with zero mean and variance of $\sigma_\epsilon^2 = \sigma^2$.

In the linear regression, it is easy to find that

$$\beta_1 = \frac{\sum_i (x_i - \bar{x})(y_i - \bar{y})}{\sum_i (x_i - \bar{x})^2}. \tag{A2}$$

where $\bar{x}$ and $\bar{y}$ are the means of $x$ and $y$ in the segment, respectively. Since

$$\sum_i (x_i - \bar{x})(y_i - \bar{y}) = \sum_i (x_i - \bar{x})y_i - \sum_i (x_i - \bar{x})\bar{y} \tag{A3}$$

and

$$\sum_i (x_i - \bar{x})\bar{y} = \bar{y}\left(\sum_i x_i - N\bar{x}\right) = \bar{y}(N\bar{x} - N\bar{x}) = 0, \tag{A4}$$

we have

$$\beta_1 = \frac{\sum_i (x_i - \bar{x}) y_i}{\sum_i (x_i - \bar{x})^2}, \tag{A5}$$

where $N$ is the number of the gates in the segment.

It is noted that the range gate $r$ is equally spaced with an interval of $\Delta r$, $\Psi_{dp}$ is the two-way propagation phase shift, and
5  $K_{dp}$ is the one-way specific differential phase. The $K_{dp}$ is then estimated by

$$K_{dp} = \frac{\sum_{i=1}^{n} \Psi_{dp}(r_i) \left[ 6i - 3(n+1)\Delta r \right]}{n(n-1)(n+1)\Delta r^2}. \tag{A6}$$

At S-band, the backscattering differential phase shift $\delta_{co}$ is often negligible, and thus $\Psi_{dp}$ and $\Phi_{dp}$ are interchangeable, leading to Eq. (2).

By taking the variance on both sides of Eq. (A5) and noting $\epsilon$ is the only variable, we have

$$10 \quad \sigma^2(\beta_1) = \sigma^2 \left( \frac{\sum_i (x_i - \bar{x})(\beta_0 + \beta_1 x_i + \epsilon)}{\sum_i (x_i - \bar{x})^2} \right) \tag{A7}$$

$$= \frac{\sum_i (x_i - \bar{x})^2 \sigma_\epsilon^2}{\left[ \sum_i (x_i - \bar{x})^2 \right]^2} \tag{A8}$$

$$= \frac{\sigma^2}{\sum_i (x_i - \bar{x})^2} \tag{A9}$$

Similar to Eq. (A6), we have

$$\sigma^2(K_{dp}) = \frac{3\sigma^2(\Psi_{dp})}{\Delta r^2 \left[ n(n-1)(n+1) \right]}. \quad \# \tag{A10}$$

15  **Appendix B: Variance of $\Phi_{dp}$**

We consider the range $r$ as an independent variable, denoted as $x$, and $\Phi_{dp}$ as a dependent variable, denote as $y$. The joint distribution of $z = (x, y)$ follows a Gaussian mixture as

$$p(z) = \sum_{i=1}^{m} w_i \mathcal{N}(z; \mu_i, \Sigma_i), \tag{B1}$$

where $w_i$, $\mu_i$ and $\Sigma_i$ are the weight, mean and covariance for each component, respectively. The probability of $y$ conditioned
20  on $x$ is also a Gaussian mixture with parameters $w_i^{y|x}$, $\mu_i^{y|x}$ and $\Sigma_i^{y|x}$, leading to the conditional expectation as

$$E(y|x) = \int y \sum_{i=1}^{m} w_i^{y|x} \mathcal{N} \left( y; \mu_i^{y|x}, \Sigma_i^{y|x} \right) dy, \tag{B2}$$

$$= \sum_{i=1}^{m} w_i^{y|x} \int y \, \mathcal{N} \left( y; \mu_i^{y|x}, \Sigma_i^{y|x} \right) dy, \tag{B3}$$

$$= \sum_{i=1}^{m} w_i \mu_i^{y|x}. \tag{B4}$$

and the second-order moment as

$$E(y^2|x) = \int y^2 \sum_{i=1}^{m} w_i^{y|x} \mathcal{N}\left(y; \mu_i^{y|x}, \Sigma_i^{y|x}\right) dy, \tag{B5}$$

$$= \sum_{i=1}^{m} w_i^{y|x} \int y^2 \mathcal{N}\left(y; \mu_i^{y|x}, \Sigma_i^{y|x}\right) dy, \tag{B6}$$

$$= \sum_{i=1}^{m} w_i^{y|x} \left[\Sigma_i^{y|x} + \left(\mu_i^{y|x}\right)^2\right]. \tag{B7}$$

Therefore, the conditional variance is expressed as

$$\sigma^2(y|x) = E(y^2|x) - [E(y|x)]^2 \tag{B8}$$

$$= \sum_{i=1}^{m} w_i^{y|x} \left[\Sigma_i^{y|x} + \left(\mu_i^{y|x}\right)^2\right] - \left(\sum_{i=1}^{m} w_i^{y|x} \mu_i^{y|x}\right)^2. \quad \# \tag{B9}$$

## Appendix C: Conditional expectation of $K_{dp}$

First, we need to show that the derivative of the expected value of random variable $y$ as a function of random variable $x$ is equal to the expected value of the derivative of the expected value of $y$. By the definition, the derivative of $y$ is expressed as

$$E'[y(x)] = \lim_{h \to 0} \frac{1}{h} \{E[y(x+h)] - E[y(x)]\} \tag{C1}$$

$$= \lim_{h \to 0} E\left[\frac{y(x+h) - y(x)}{h}\right] \tag{C2}$$

$$= \lim_{h \to 0} E\{y'[\tau(h)]\}, \tag{C3}$$

where $\tau(h) \in (x, x+h)$ exists by the mean value theorem. By assuming $|y'[\tau(h)]| \leq Z$, we can use the dominated convergence theorem to obtain

$$E'[y(x)] = E\left\{\lim_{h \to 0} y'[\tau(h)]\right\} \tag{C4}$$

$$= E[y'(x)]. \tag{C5}$$

According to the conclusion in Eq. (C5), the expected value of $y'$ is expressed as

$$E(y'|x) = E'(y|x) = \sum_{i=1}^{m} w_i^{y|x} \left(\mu_i^{y|x}\right)' + \sum_{i=1}^{m} \left(w_i^{y|x}\right)' \mu_i^{y|x}. \tag{C6}$$

Since $\left(\mu_i^{y|x}\right)' = a_i$ and $w_i^{y|x} = \frac{f_i(x)}{f(x)}$, the first term is equal to

$$\sum_{i=1}^{m} w_i^{y|x} \left(\mu_i^{y|x}\right)' = \sum_{i=1}^{m} a_i \frac{f_i(x)}{f(x)}. \tag{C7}$$

Meanwhile, the second term is given as

$$\sum_{i=1}^{m} \left( w_i^{y|x} \right)' \mu_i^{y|x} = \sum_{i=1}^{m} \frac{f_i'(x)f(x) - f_i(x)f'(x)}{f^2(x)} (a_i x + b_i). \tag{C8}$$

Based on the properties of Gaussian function, the derivatives of $f_i(x)$ and $f(x)$ are expressed as

$$f_i'(x) = -\frac{x - \mu_i^x}{\Sigma_i^{xx}} f_i(x), \quad \text{and} \tag{C9}$$

$$f'(x) = -\sum_{j=1}^{m} \frac{x - \mu_j^x}{\Sigma_j^{xx}} f_j(x). \tag{C10}$$

By substituting Eqs. (C9) and (C10) into Eq. (C8), the second term is transformed as

$$\sum_{i=1}^{m} \left( w_i^{y|x} \right)' \mu_i^{y|x} = \frac{1}{f^2(x)} \left[ \sum_{i=1}^{m} \sum_{j=1}^{m} \left( \frac{x - \mu_j^x}{\Sigma_j^{xx}} - \frac{x - \mu_i^x}{\Sigma_i^{xx}} \right) f_i(x) f_j(x) (a_i x + b_i) \right]. \tag{C11}$$

By substituting Eqs. (C7) and (C11) into Eq. (C6), we obtain

$$E\left( y' | x \right) = \frac{1}{f^2(x)} \left\{ \sum_{i=1}^{m} \sum_{j=1}^{m} f_i(x) f_j(x) \left[ \left( \frac{x - \mu_j^x}{\Sigma_j^{xx}} - \frac{x - \mu_i^x}{\Sigma_i^{xx}} \right) (a_i x + b_i) + a_i \right] \right\} \quad \#. \tag{C12}$$

## Appendix D: Variance of $K_{dp}$

The first-order Taylor expansion is defined as

$$g(y) = g(\theta) + g'(\theta)(y - \theta) + \epsilon, \tag{D1}$$

where $\theta = E(y)$ is the mean of random variable $y$, and $\epsilon$ is the sum of the higher-order Taylor series. By considering the conclusion in Eq. (C5), it can be noted that the expected values of the coefficients associated with the derivatives in Eq. (D1) are zeros if the series is expanded at the mean value of $y$. By taking mathematical expectations on both sides of Eq. (D1), it is transformed as

$$E[g(y)] \approx g(\theta) + g'(\theta)[E(y) - \theta] \tag{D2}$$

$$= g(\theta). \tag{D3}$$

From Eqs. (D1) and (D3), the variance of $g(y)$ is approximated as

$$\sigma^2[g(y)] \approx E\left\{ [g(y) - g(\theta)]^2 \right\} \tag{D4}$$

$$\approx E\left\{ [g'(\theta)(y - \theta)]^2 \right\} \tag{D5}$$

$$= g'(\theta)^2 \sigma^2(y) \tag{D6}$$

Let $g(y)$ be $y'$, and then we have

$$\sigma^2(y'|x) = [E''(y|x)]^2 \sigma^2(y|x). \tag{D7}$$

From Eq. (B9), we can see that

$$\sigma^2(y|x) = \sum_{i=1}^{m} w_i^{y|x} \Sigma_i^{y|x} + \sum_{i=1}^{m} w_i^{y|x} \left(\mu_i^{y|x}\right)^2 - \left(\sum_{i=1}^{m} w_i^{y|x} \mu_i^{y|x}\right)^2. \tag{D8}$$

By taking the derivative of Eq. (C6), the second derivative of the expected value of $y$ conditioned on $x$ becomes

$$E''(y|x) = \sum_{i=1}^{m} w_i^{y|x} (\mu_i^{y|x})'' + 2\sum_{i=1}^{m} (\mu_i^{y|x})' (w_i^{y|x})' + \sum_{i=1}^{m} \mu_i^{y|x} (w_i^{y|x})'' \tag{D9}$$

$$= 2\sum_{i=1}^{m} a_i (w_i^{y|x})' + \sum_{i=1}^{m} (a_i x + b_i)(w_i^{y|x})'', \tag{D10}$$

since $(\mu_i^{y|x})' = a_i$ and $(\mu_i^{y|x})'' = 0$. From Eq. (C8), the first derivative of the weighting function in the conditional probability is

$$\left(w_i^{y|x}\right)' = \frac{1}{f^2(x)} \sum_{j=1}^{m} f_j(x) f_i(x) \left(\frac{x - \mu_j^x}{\Sigma_j^{xx}} - \frac{x - \mu_i^x}{\Sigma_i^{xx}}\right). \tag{D11}$$

Let $g_i(x)$ be the summation term. The second derivative is then expressed as

$$\left(w_i^{y|x}\right)'' = \frac{g_i'(x) f(x) - 2f'(x) g_i(x)}{f^3(x)}, \quad \text{where} \tag{D12}$$

$$g_i(x) = \sum_{j=1}^{m} f_j(x) f_i(x) \left(\frac{x - \mu_j^x}{\Sigma_j^{xx}} - \frac{x - \mu_i^x}{\Sigma_i^{xx}}\right). \tag{D13}$$

According to the properties of Gaussian mixture, the first derivative of the marginal distribution of $x$ is

$$f'(x) = -\sum_{j=1}^{m} \left(\frac{x - \mu_j}{\Sigma_j}\right) f_j(x). \tag{D14}$$

Similarly, the first derivative of $g(x)$ if given as

$$g_i'(x) = \sum_{j=1}^{m} \left[ -\frac{x - \mu_j^x}{\Sigma_j^{xx}} \left(\frac{x - \mu_j^x}{\Sigma_j^{xx}} - \frac{x - \mu_i^x}{\Sigma_i^{xx}}\right) f_j(x) f_i(x) - \frac{x - \mu_i^x}{\Sigma_i^{xx}} \left(\frac{x - \mu_j^x}{\Sigma_j^{xx}} - \frac{x - \mu_i^x}{\Sigma_i^{xx}}\right) f_j(x) f_i(x) + \left(\frac{1}{\Sigma_j^{xx}} - \frac{1}{\Sigma_i^{xx}}\right) f_j(x) f_i(x) \right] \tag{D15}$$

$$= \sum_{j=1}^{m} \left[ \left(\frac{x - \mu_i^x}{\Sigma_i^{xx}} - \frac{x - \mu_j^x}{\Sigma_j^{xx}}\right) \left(\frac{x - \mu_i^x}{\Sigma_i^{xx}} + \frac{x - \mu_j^x}{\Sigma_j^{xx}}\right) f_j(x) f_i(x) + \left(\frac{1}{\Sigma_j^{xx}} - \frac{1}{\Sigma_i^{xx}}\right) f_j(x) f_i(x) \right] \tag{D16}$$

$$= \sum_{j=1}^{m} \left[ \left(\frac{x - \mu_i^x}{\Sigma_i^{xx}}\right)^2 - \left(\frac{x - \mu_j^x}{\Sigma_j^{xx}}\right)^2 \right] f_j(x) f_i(x) + \left(\frac{1}{\Sigma_j^{xx}} - \frac{1}{\Sigma_i^{xx}}\right) f_j(x) f_i(x) \tag{D17}$$

$$= \sum_{j=1}^{m} f_j(x) f_i(x) \left[ \left(\frac{x - \mu_i^x}{\Sigma_i^{xx}}\right)^2 - \left(\frac{x - \mu_j^x}{\Sigma_j^{xx}}\right)^2 + \frac{1}{\Sigma_j^{xx}} - \frac{1}{\Sigma_i^{xx}} \right]. \quad \# \tag{D18}$$

*Data availability.* The MZZU radar data can be made available upon request to the authors. The rain gauge data are available online: http://agebb.missouri.edu/weather/stations/index.php.

*Author contributions.* NF designed the experiment and provided the radar data, GW developed the Gaussian mixture model and prepared the manuscript, GW and NF performed the validation, and NF and PM reviewed the paper.

5   *Competing interests.* The authors declare that they have no conflict of interest

*Acknowledgements.* The authors would like to express our sincere thanks to the anonymous reviewers for their valuable comments and suggestions. This work was supported by Missouri Experimental Project to Stimulate Competitive Research (EPSCoR) of National Science Foundation, under Award Number IIA-1355406. Any opinions, findings, and conclusions or recommendations expressed in this material are those of the authors and do not necessarily reflect the views of the National Science Foundation.

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

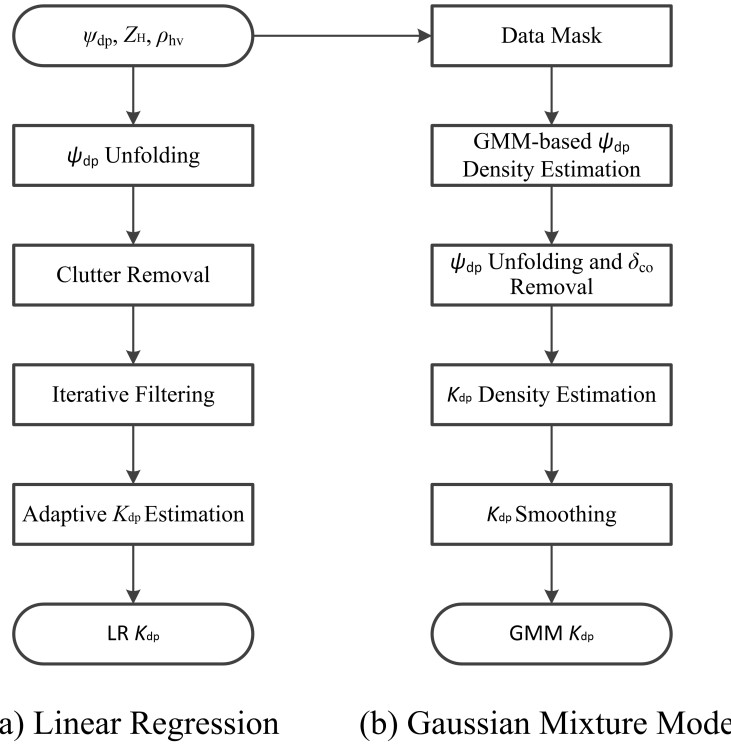

(a) Linear Regression     (b) Gaussian Mixture Model

**Figure 1.** Flowcharts of $K_{dp}$ estimation algorithms used in the MZZU radar: (a) linear regression model, and (b) Gaussian mixture method.

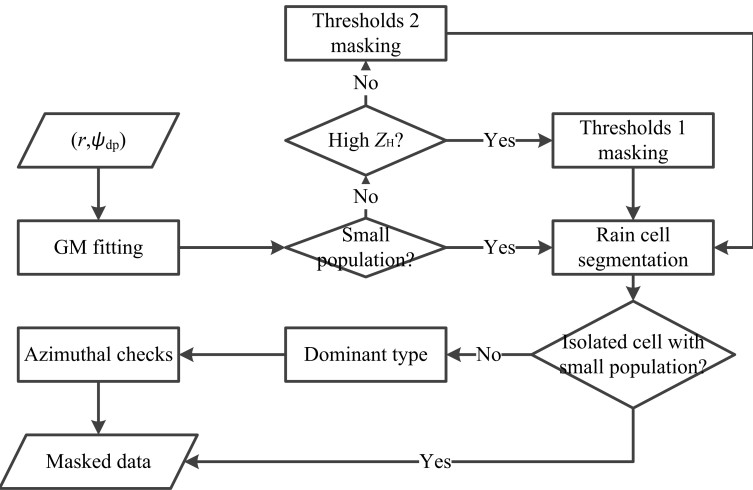

**Figure 2.** Flowchart of data masking.

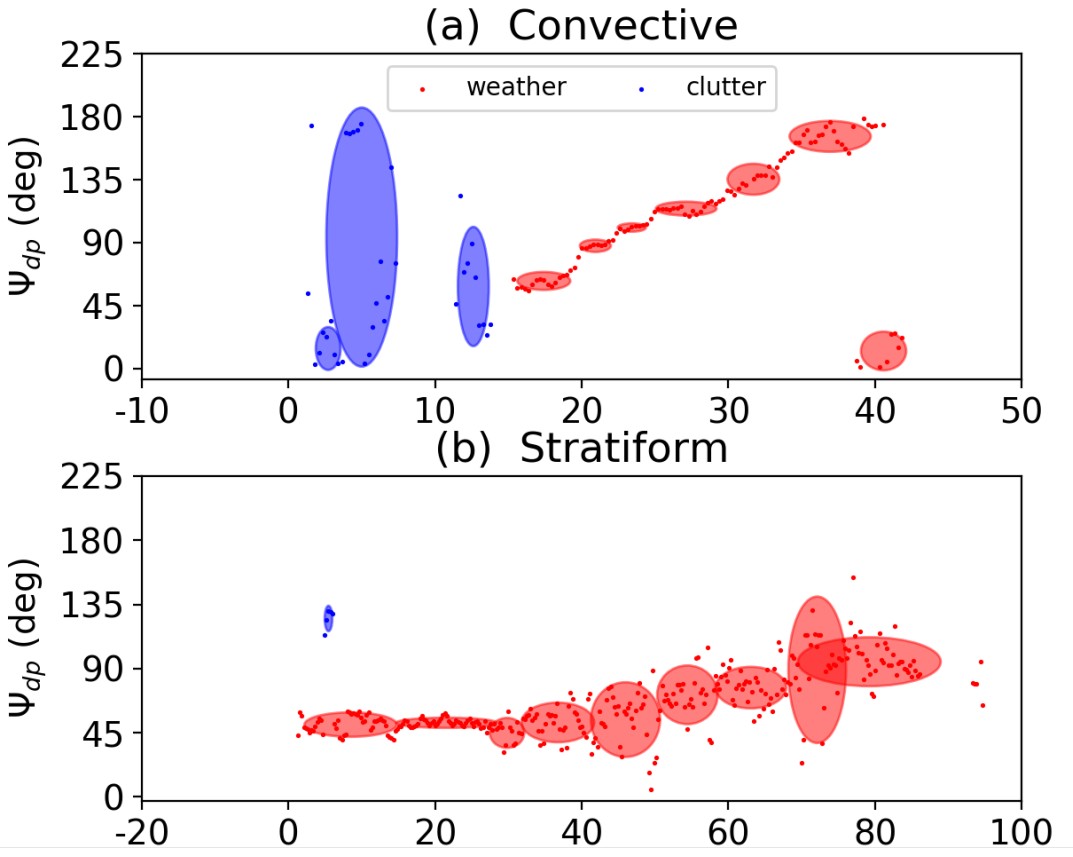

**Figure 3.** Examples of data masking: (a) a convective case (azimuth $252°$), and (b) a stratiform case (azimuth $1°$). The blue points and ellipses represent the clutter data and clusters, respectively, while the red color corresponds to the weather echoes. The x-axis is the radar range in km.

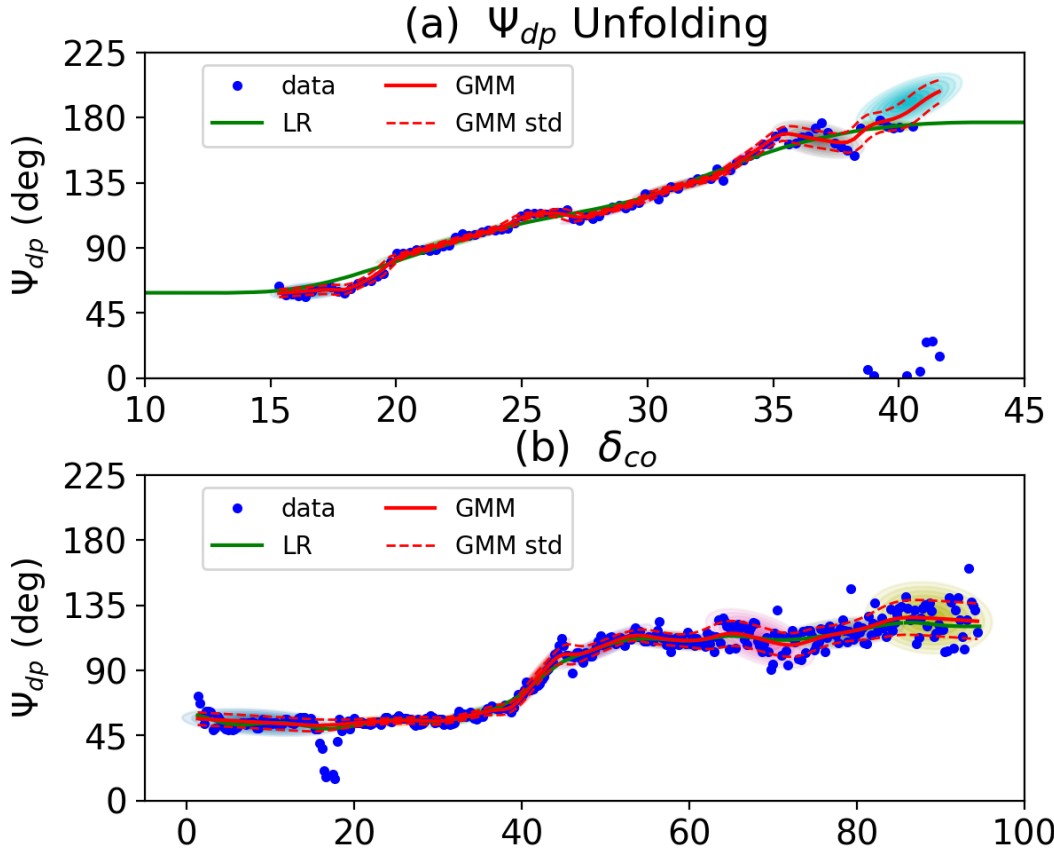

**Figure 4.** Examples of $\Psi_{dp}$ density estimation: (a) a $\Psi_{dp}$ unfolding case, and (b) a $\delta_{co}$ case. The blue points are the $\Psi_{dp}$ data, the green curve represents the $\Psi_{dp}$ profile obtained by the linear regression model (LR), and the red curve indicates the $\Psi_{dp}$ profile produced by the Gaussian mixture method (GMM). The dash lines are the standard deviations, while the colored ellipses show the components of the Gaussian mixture. The x-axis is the radar range in km.

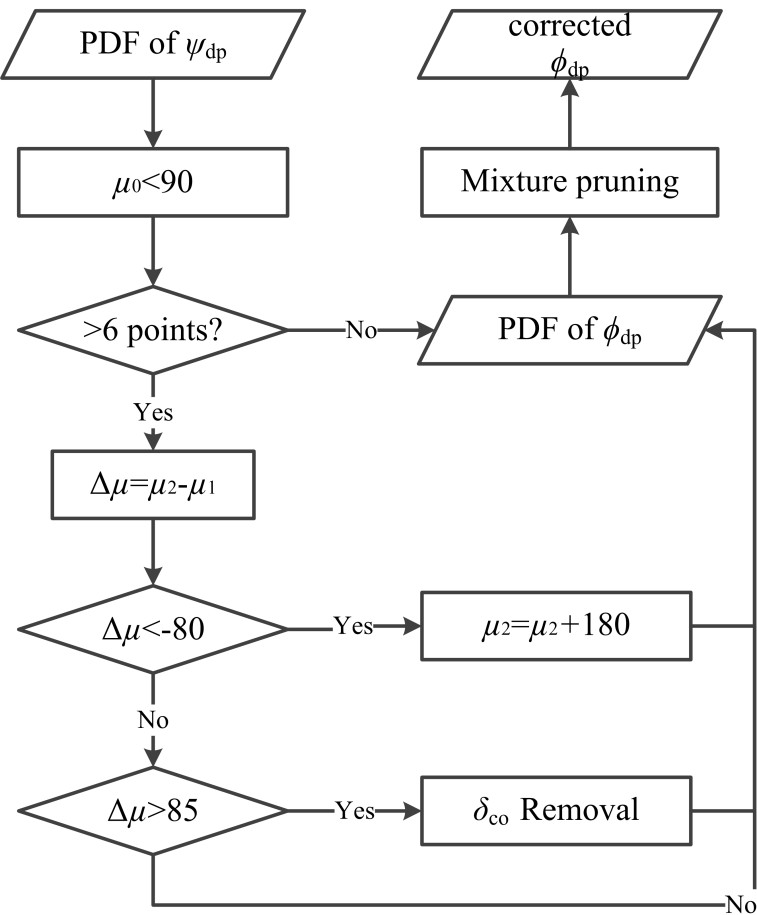

**Figure 5.** Flowchart of the $\Psi_{dp}$ unfolding and the $\delta_{co}$ elimination. The $\mu_0$ is the mean of the first density ellipse. The $\mu_1$ and $\mu_2$ are the means of the two consective density ellipses along the range. The $\mu_1$ is the mean of the former one, and the $\mu_2$ is the mean of the latter one.

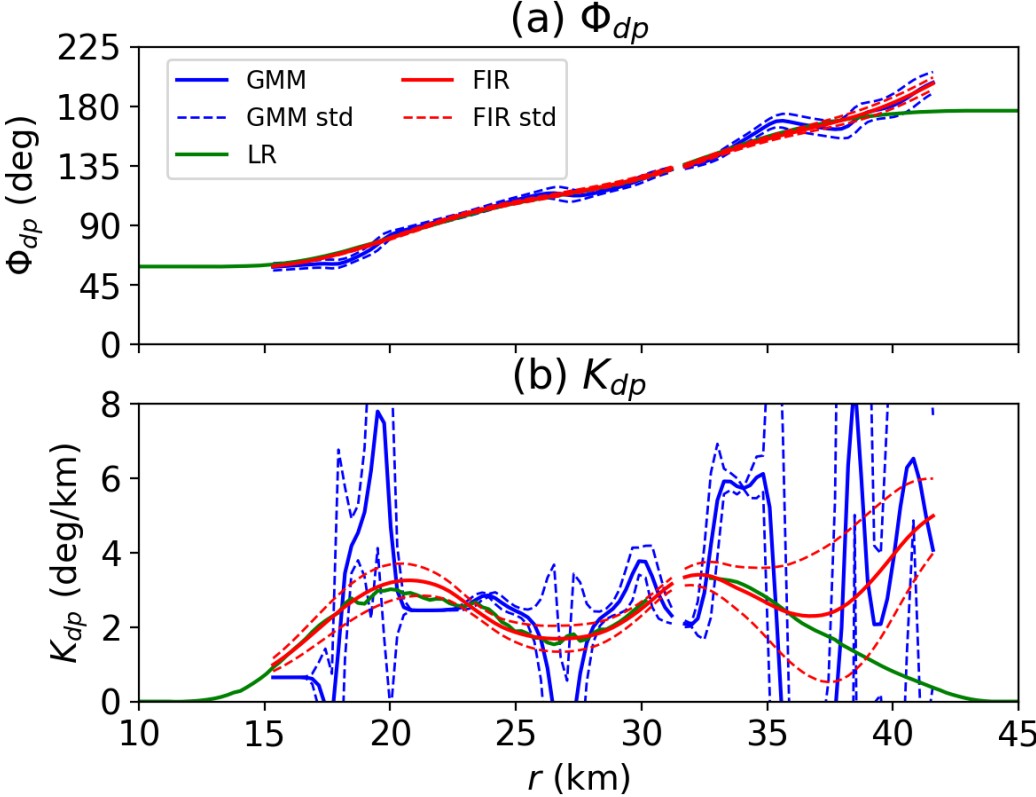

**Figure 6.** Examples of $K_{dp}$ estimation: (a) $\Phi_{dp}$, and (b) $K_{dp}$. The blue curves are the $\Phi_{dp}$ and $K_{dp}$ estimates obtained by the Gaussian mixture method (GMM), the green curves represent the estimates derived from the linear regression model (LR), and the red curves indicate the reconstructed $\Phi_{dp}$ and smoothed $K_{dp}$ profiles (FIR). The dash lines are the standard deviations.

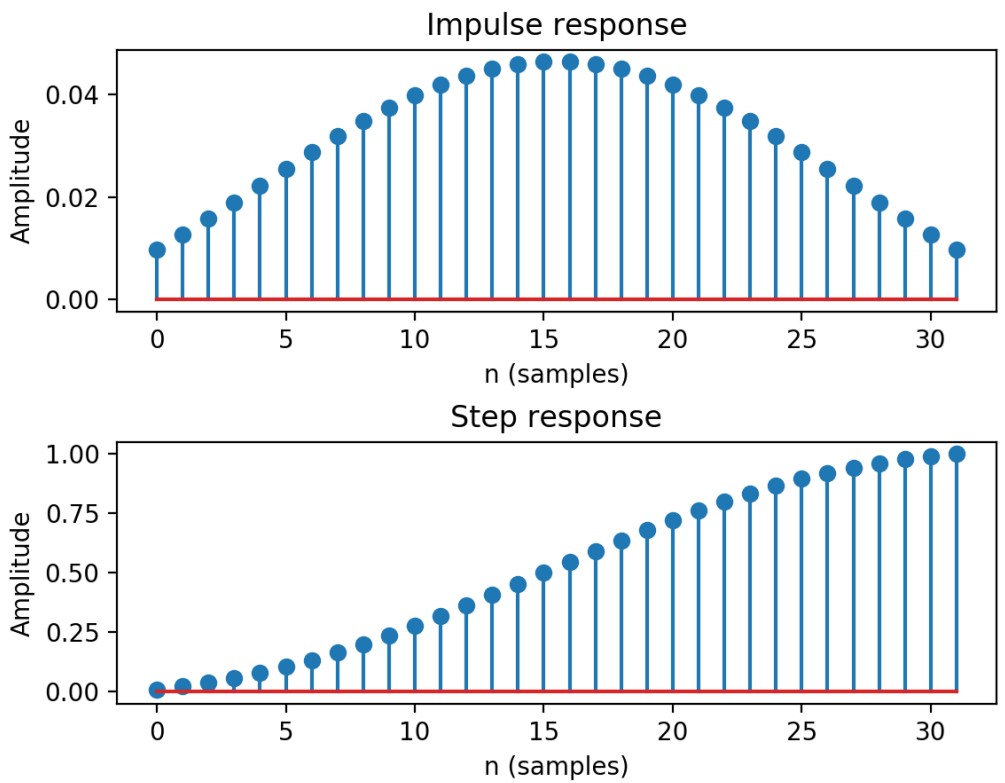

**Figure 7.** Responses of finite impulse filter: (a) Impulse response, and (b) step response.

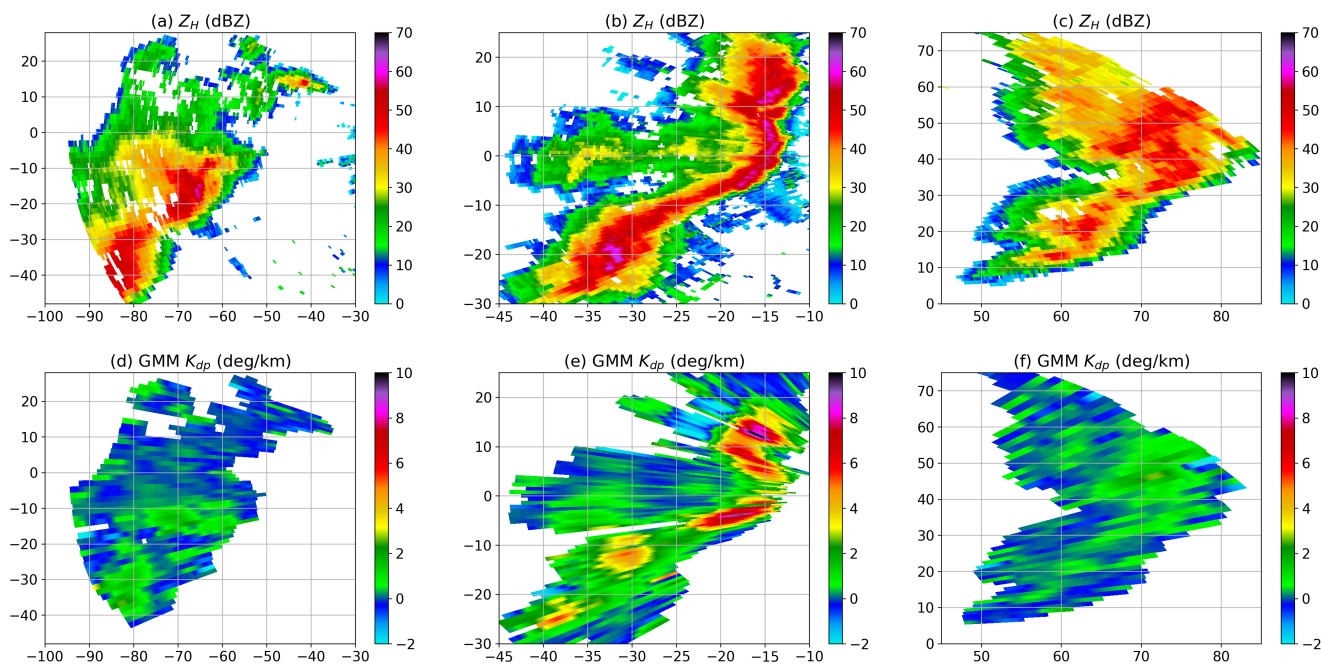

**Figure 8.** A case study for GMM: (a) raw $Z_H$ at the development stage (03:04 UTC), (b) raw $Z_H$ at the mature stage (03:39 UTC), and (c) raw $Z_H$ at the dissipation stage (04:41 UTC). (d), (e) and (f) are the same as (a), (b) and (c), respectively, but for $K_{dp}$. The data were collected at a elevation of $0.85°$ by the MZZU radar between 0304 and 0441 UTC on 24 March 2016.

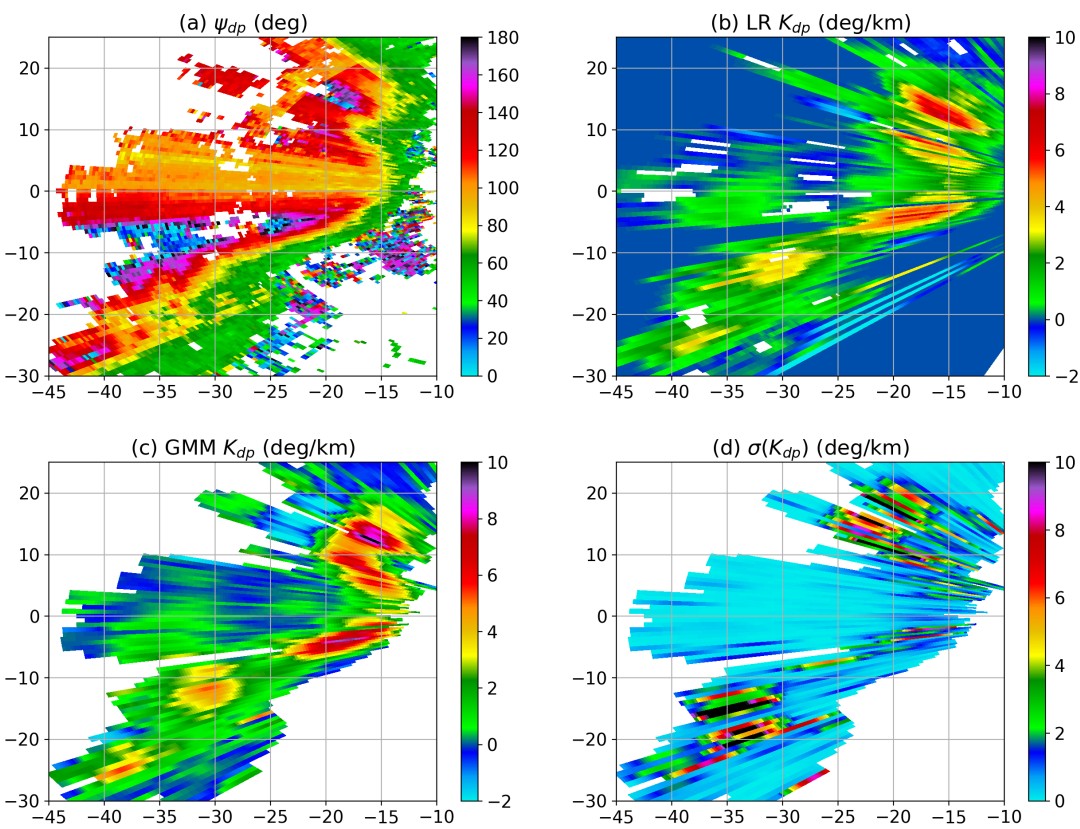

**Figure 9.** $K_{dp}$ estimation for the mature stage: (a) raw $\Psi_{dp}$, (b) LR-based $K_{dp}$, (c) GMM-based $K_{dp}$, and (d) GMM-based $\sigma(K_{dp})$. The data were collected at 0339 UTC on 24 March 2016.

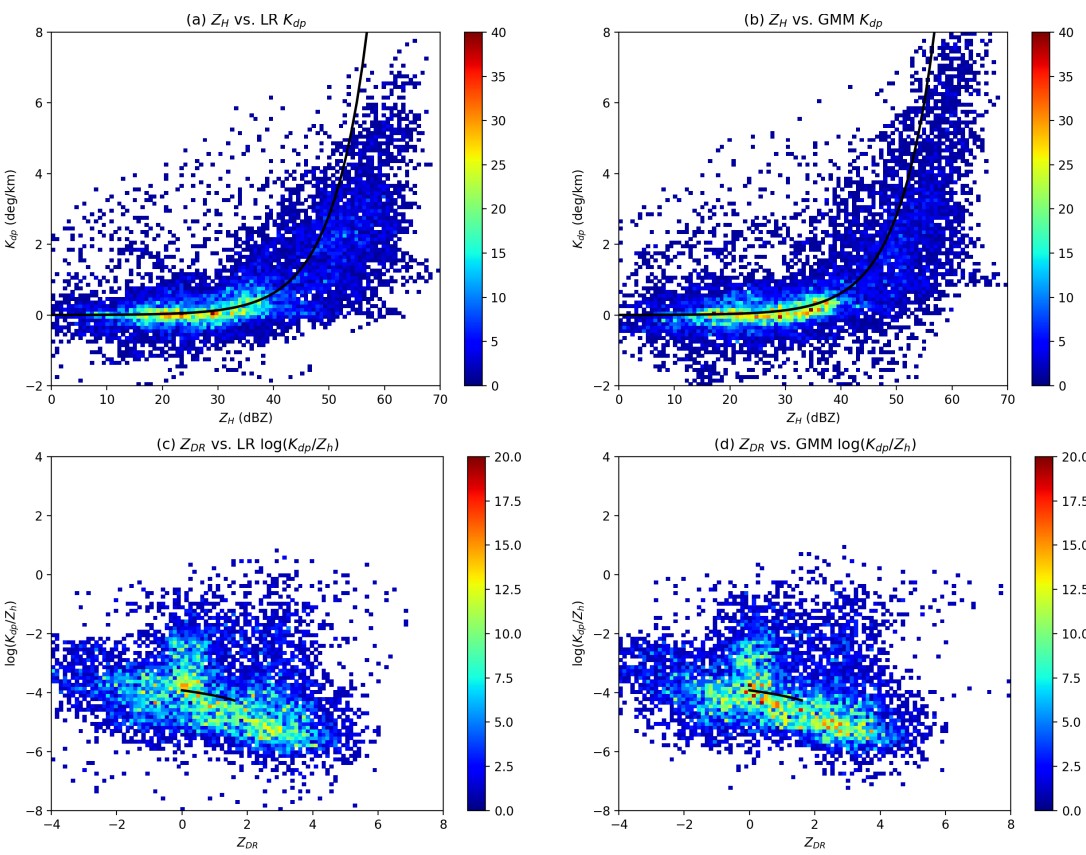

**Figure 10.** Comparison with the self-consistency relations: (a) $Z_H$ vs. LR $K_{dp}$, (b) $Z_H$ vs. GMM $K_{dp}$, (b) $Z_{DR}$ vs. the ratio of LR $K_{dp}$ and $Z_h$ and (b) $Z_{DR}$ vs. the ratio of GMM $K_{dp}$ and $Z_h$, where $Z_H = 10 \log(Z_h)$. The color scale is the number of points, and the black curves are the theoretical self-consistency relations.

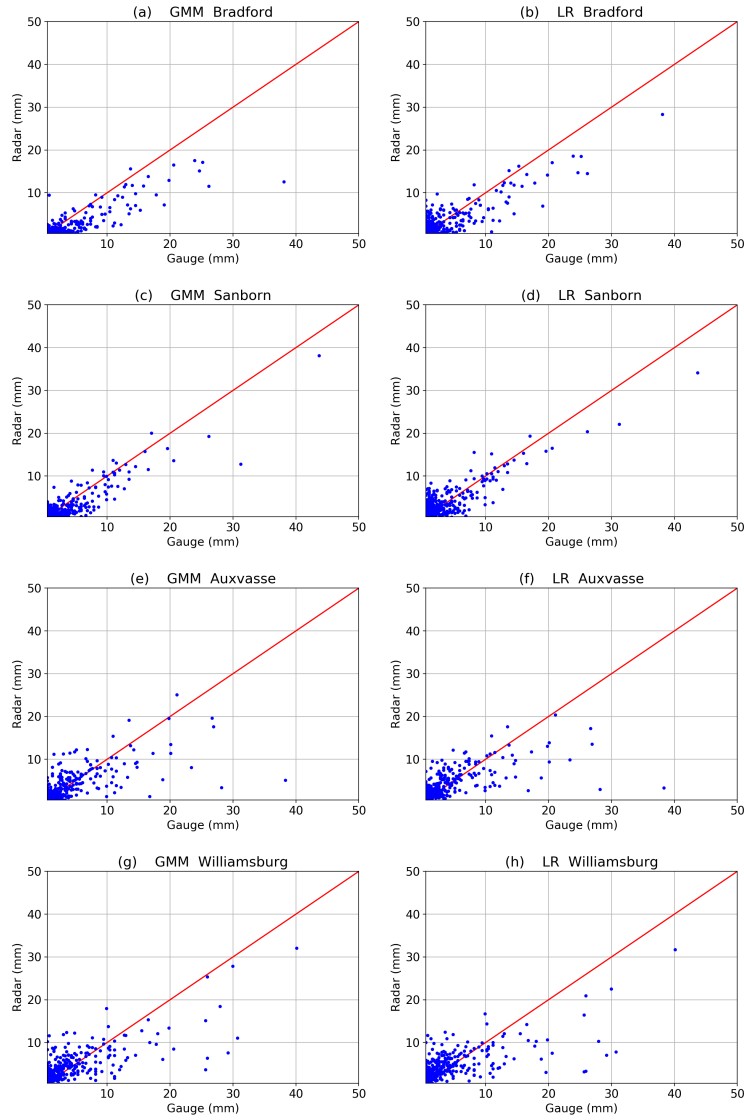

**Figure 11.** Comparison between hourly radar and gauge data derived from GMM $K_{dp}$ and LR $K_{dp}$. (a) GMM Bradford, (b) LR Bradford, (c) GMM Sanborn, (d) LR Sanborn, (e) GMM Auxvasse, (f) LR Auxvasse, (g) GMM Williamsburg and (h) LR Williamsburg. The data were collected between 1 April 2016 and 2 June 2018.

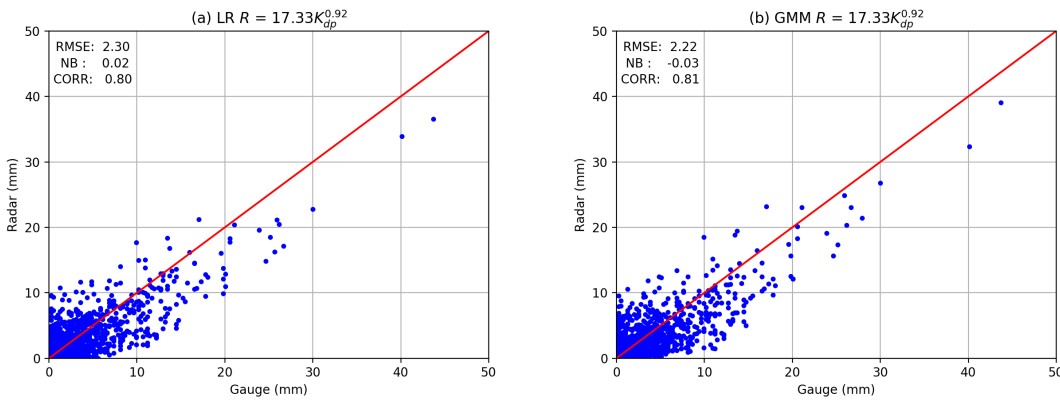

**Figure 12.** Same as Fig. 11, but for the optimal $R$-$K_{dp}$ relation for the four sites. (a) LR $R(K_{dp})$ and (b) GMM $R(K_{dp})$.

**Table 1.** Characteristics of hourly rain gauge data at Bradford, Sanborn, Auxvasse, and Williamsburg between April 2016 and June 2018. Mean: mean values, Std: standard deviation, Max: maximum values, Total: sums of rain amounts, and Duration: sum of rainfall time.

| Sites | Mean (mm) | Std (mm) | Max (mm) | Total (mm) | Duration (h) |
|---|---|---|---|---|---|
| Bradford | 2.1 | 3.5 | 38.1 | 2224.9 | 1080 |
| Sanborn | 2.0 | 3.3 | 43.7 | 2181.4 | 1082 |
| Auxvasse | 2.0 | 3.3 | 38.4 | 2284.3 | 1144 |
| Williamsburg | 2.1 | 3.7 | 40.1 | 2495.9 | 1191 |

**Table 2.** Statistics for the comparison between radar and gauge. RMSE: root mean squared error, NB: normalized bias, $\rho_{RG}$: Pearson correlation coefficient; LR: linear regression model, GMM: Gaussian mixture method.

| Algorithm | Sites | RMSE (mm) | NB | $\rho_{RG}$ |
|-----------|-------|-----------|-----|------|
| LR | Bradford | 2.87 | -0.28 | 0.84 |
| | Sanborn | 1.97 | -0.08 | 0.89 |
| | Auxvasse | 3.25 | 0.21 | 0.67 |
| | Williamsburg | 3.55 | 0.20 | 0.70 |
| GMM | Bradford | 2.71 | -0.31 | 0.84 |
| | Sanborn | 2.06 | -0.13 | 0.88 |
| | Auxvasse | 3.14 | 0.04 | 0.69 |
| | Williamsburg | 3.20 | 0.14 | 0.76 |