# Peer review of "A Gaussian Mixture Method for Specific Differential Phase Retrieval at X-band Frequency"

_Atmospheric Measurement Techniques, 2019_

## Referee Comment (RC1) · Anonymous Referee #1 · 8 Jul 2019

I have carefully, and with interest, read the manuscript. The paper addresses estimation of the Kdp based on the measurements of the differential phase shift in X-band polarimetric radars. The authors use data from the University of Missouri X-band radar.

Overall, I find the paper to be technically sound and worth publishing. The key comments are:

Readers would benefit from a more tutorial style as the topic is highly specialized. This regards both the Kdp estimation in general as well as the Gaussian mixture statistical modeling.

I think, that it is important to point out that Kdp is calculated from a filtered (estimated) differential phase and not directly from its moment-based measurements. To distin-

guish the three, one may use psi, fi, k symbols.

The authors should emphasize that the main advantage of their proposed method is in providing the estimation variance for the Kdp and not is providing better estimates of Kdp. This is evident in the long-term evaluation using rain gauge data.

I suggest that the authors improve the quality of the figures: some lettering is not legible, the inter-panel space could be reduced, etc.
* * *

---

## Referee Comment (RC2) · Anonymous Referee #2 · 10 Jul 2019

I have found the paper to be very interesting and quite well written and structured. It proposes a novel approach (Gaussian Mixture Regression) to a classical problem ( $K_{dp}$  retrieval). Technically and mathematically I found it very sound and I think it is worth publishing. Its main contribution is the possibility to give an error estimate to the retrieved specific differential phase, which is still uncommon. The method is quite complicated and as such it would be amazing if the authors considered implementing it in a radar toolkit such as for example Py-ART https://arm-doe.github.io/pyart/, which already contains a few retrieval methods.

**Major comments**

- 1. Section 4.3 and later : I think that you should use a different notation for the raw measured differential phase shift on propagation and the filtered version with  $\delta_{co}$  removed. Usually the notation  $\psi_{dp}$  is used for the raw measurement and  $\phi_{dp}$  for the filtered signal from which  $K_{dp}$  is estimated.
- 2. One major issue in an operational context is the computational cost of these more sophisticated  $K_{dp}$  retrieval techniques. Mainly for this reason, the standard linear regression methods are still the norm. Could you discuss and provide numbers for the computational cost of your method and maybe compare it with other methods?
- 3. In the conclusion, I think it would be interesting to discuss if this method could be used as such for other frequencies (C-band and S-band in particular) or if it would require some relevant tweaks.

**Minor comments**

- 1. p2. l.11-12 : This sentence is not very clear and syntactically correct, please reformulate
- 2. p2 : I.23 : Like the proposed method, the Kalman filter method also provides an estimate of the standard deviation of the retrieved KDP at X-band, it would be interesting to explain it in in broader detail as well as discuss the differences and respective advantages of both methods.
- 3. p11: I.3-5: I have trouble understanding this paragraph. I would suggest to reformulate to make it clearer, in particular the term "transformed into the next stage" is inappropriate.
- 4. I would suggest to add another flowchart for the step  $\phi_{dp}$  unfolding and  $\delta_{co}$  estimation.
- 5. p.13 l.17-18: It would be good to discuss why you choose this particular FIR filter. I am also not sure how the number of considered gates is defined.
- 6. p.16 l.8: It would be good to include one or two sentences that explain briefly this X-band rainfall rate algorithm.
- 7. Figures 4 and 5 should be visually improved. In particular the data points are too hard to see because of the error bars. I would for example replace the error bars by thin lines located one each side of the plot. Also the limits of the y axis could be adjusted.
- 8. Figure 9: It would be useful to also include the radar estimates derived from the LR  $K_{dp}$ .

---

## Referee Comment (RC3) · Anonymous Referee #3 · 16 Jul 2019

Estimating Kdp is a quite difficult task in radar meteorology, being estimation based on computin a derivative of a range profile affected by noise and various measurement errors. Different methods exist, but are difficult to compare each other, since all comes with advantages and disadvantages. The manuscript by Wen et al. describes a novel approach, called Gaussian Mixture Method (GMM) is proposed and compare its performance with the classical method based on linear regression (LR). Results in terms of QPE computed with respect to 4 raingauges at different distance from the radar shows a slightly better (but not for all the raingauges, only for the two farther from the radar) performance of the new method. It is not easy to provide a convincing validation of Kdp estimation, but some suggestions to improve it are provided below (note than pages 2-14 describes the method, but only pages 14-16 are related to validation).

[Figure]

More difficult is to get an ultimate proof of the effectiveness of the method, but only a set of clues. The paper is correctly structured, but needs some improvement in order to be accepted for publication. The acceptance of the method will be depend also on practical aspects, such as the computational effort and robustness that, unfortunately are not considered in this manuscript. In the following, general and specific issues are listed.

General comments:

There is a problem in the use of symbols that affect the clarity of the manuscript. Kdp is estimated from the range derivative of differential propagation phase $\Phi$dp, whereas the differential phase (usually indicated with $\Psi$dp)that includes the differential phase upon backscattering, that is not negligible at X band) is measured by radar. Moreover, in a paper about estimation methods, the estimated variable should be clearly distinct from the intrinsic variable. Usually, estimated variables are indicated with a hat on top. Adopting these standard symbols (or another set of symbols authors could prefer) would increase the readability of the manuscript.

Aspects related to computational cost are neglected. Some figures about the computational efficiency of the method need to be provided.

Introduction is quite confusing. It recall several Kdp estimation methods (some of them are missing, such as Vulpiani et. al. 2012, https://doi.org/10.1175/JAMC-D-10-05024.1), but mixes general Kdp estimation methods with other methods based on self-consistency of dual polarization measurements that are valid strictly in rain. One of the advantages of GMM is that it can provide the variance associated the Kdp estimate. I think the authors should explain why such result is important and how it can be used. Other methods, such as Kalman filter can provide the variance of the estimate, but even linear regression can provide a standard error as the measure of the goodness of fitting. The manuscript states that $\sigma(\Phi dp)$ is constant. However, notoriously $\sigma(\Phi dp)$ varies depending on SNR, width of Doppler spectrum and copolar correlation

coefficients that are not constant along the range.

Specific issues:

Page 1: Line 10, raingauge data used for validation are relative to two years and 3 months, not three years.

Page 1: Line 15, please use different symbols for the measure differential phase shift and its component related to propagation.

Page 2: Line 5-6, "Therefore...". This is something to be demonstrated, it is not a consequence of the previous statement.

Page 5: Line 25-30: Is there an implication of this sentence for Kdp estimation ?

Page 6: Line 10. What is "the maximum detectable range" ? is it the unambiguous range determined by the selected PRF ?

Page 6: Line 10-25: are the height above the ground or ASL ? for the elevation of 0.8°, at 4.4 km from radar the height of radar beam is 314.6 m. Is it correct ?

Page 8: Line 1-2: There is something odds in the sentence. Are clutter-contaminated echoes well identified or not ?

Page 8: lines 9-11. What is the need of eliminate hail contamination ? Is not Kdp computed in hail ?

Page 8: Lines 18-32: I recommend to use degrees consistently in the manuscript and add a legend on x-axis (this is valid for all the figures showing profiles). Given the range of unambiguous differential phase, it seems that MZZU features the alternate polarization scheme. About the jump at the beginning of the profile in Fig. 3b: to be interpreted as ïA̧ďco there should a peak of Zdr. Is this the case ?

Page 9. Lines 1-2: negative Kdp can be due also to non uniform beam filling (Ryzkhov 2007 https://doi.org/10.1175/JTECH2003.1), which is a further source of error in computing Kdp (see manuscript at pag 15).

Page 10. Again, please be sure that the interpretation of differential phase upon backscattering is correct.

Page 11. Line 29. Section explains how unfolding works. Sometimes there are false alarms in unfolding. Could you provide the rate of correct unfolding for the dataset used?

Page 13, line 22-23. Could authors please provide more details on how the weights are derived?

Page 14: Line 23. Please provide information about how hourly rain is obtained from instantaneous radar measurements.

Page 15: Line 13-24. The presence of too many missing data for LR indicates a clear advantage for GMM. However, it is not clear what is the cause of these missing data. Surprisingly is the presence of Kdp estimates beyond the edge of differential phase shift rays, especially for LR. Is it the effect of smoothing and/or extrapolation?

Page 15: Line 25. Notoriously, in rain, Kdp, Zh, and Zdr exhibit a self-consistency property. I recommend authors to exploit this property to validate their method to see whether the Kdp estimate is consistent with Zh and Zdr measurements.

Page 32: Figure 7: Please use larger font size.

Page 34: A Kdp rain algorithm is supposed to work well (at least better than a R(Z) algorithm) for high rainfall rate. Instead, there are evident underestimates for high rainfall rates that likely affect RMSE. How about LR: does it yield the same behavior?

Page 36: NB is not in mm.

---

## Author Comment (AC1) · 19 Jul 2019

**A Gaussian Mixture Method for Specific Differential Phase Retrieval at X-band Frequency**

Guang Wen1, Neil I. Fox1,\*, and Patrick S. Market1

1School of Natural Resources, University of Missouri, 332 ABNR Building, Columbia, Missouri, USA, 65201 **Correspondence:** Neil I. Fox (foxn@missouri.edu)

We would like to express our sincere thanks to the Editor for the efficient review management, and to the anonymous reviewers for their valuable comments and suggestions. We have addressed all the reviewers' comments point by point in the revision.

**1 Review #1**

5 1.1. Readers would benefit from a more tutorial style as the topic is highly specialized. This regards both the Kdp estimation in general as well as the Gaussian mixture statistical modeling.

[response] This is really a good comment, since the paper describes a new  $K_{dp}$  estimation method. It is necessary to give some details about this topic to readers in a variety of backgrounds. Therefore, we have provided an additional appendix related to the regression-based estimation of  $K_{dp}$ .

10 [changes] p.4, ln.10–12: This method has been widely used in the existing radar system (Cifelli et al., 2018; Chandrasekar et al., 2018; Chen et al., 2017b, a). The details of the regression-based estimation of  $K_{dp}$  are given in Bringi and Chandrasekar (2001) and Appendix A.

[changes] p.19, ln.6–p.20, ln.6: Appendix A: Regression-based estimation of  $K_{dp}$

Let the total differential phase  $\psi_{dp}$  be y, and the range gate r be x. The  $\psi_{dp}$  profile over small range segments can be 15 approximated by a first-order polynomial, i.e,i

$$y = \beta_0 + \beta_1 x + \epsilon, \tag{1}$$

where  $\beta_0$  and  $\beta_1$  are the coefficients in the linear approximation, and  $\epsilon$  is an error function. It can be assumed that  $\epsilon$  is independent and individual distributed with zero mean and variance of  $\sigma_{\epsilon}^2 = \sigma^2$ .

In the linear regression, it is easy to find that

20
$$\beta_1 = \frac{\sum_i (x_i - \bar{x})(y_i - \bar{y})}{\sum_i (x_i - \bar{x})^2}.$$
 (2)

where  $\bar{x}$  and  $\bar{y}$  are the means of x and y in the segment, respectively. Since

$$\sum_{i} (x_i - \bar{x})(y_i - \bar{y}) = \sum_{i} (x_i - \bar{x})y_i - \sum_{i} (x_i - \bar{x})\bar{y}$$
(3)

and

$$\sum_{i} (x_i - \bar{x})\bar{y} = \bar{y}\left(\sum_{i} x_i - N\bar{x}\right) = \bar{y}(N\bar{x} - N\bar{x}) = 0,\tag{4}$$

we have

$$\beta_1 = \frac{\sum_i (x_i - \bar{x}) y_i}{\sum_i (x_i - \bar{x})^2},$$
(5)

5 where N is the number of the gates in the segment.

It is noted that the range gate r is equally spaced with an interval of  $\Delta r$ ,  $\psi_{dp}$  is the two-way propagation phase shift, and  $K_{dp}$  is the one-way specific differential phase. The  $K_{dp}$  is then estimated by

$$K_{dp} = \frac{\sum_{i=1}^{n} \psi_{dp}(r_i) \left[ i - \frac{(n+1)}{2} \Delta r \right]}{\frac{1}{6} n(n-1)(n+1) \Delta r^2}.$$
(6)

At S-band, the backscattering differential phase shift  $\delta_{co}$  is often negligible, and thus  $\psi_{dp}$  and  $\phi_{dp}$  are interchangeable, leading 10 to Eq. (2).

By taking the variance on both sides of Eq. (5) and noting  $\epsilon$  is the only variable, we have

$$\sigma^{2}(\beta_{1}) = \sigma^{2} \left( \frac{\sum_{i} (x_{i} - \bar{x})(\beta_{0} + \beta_{1}x_{i} + \epsilon)}{\sum_{i} (x_{i} - \bar{x})^{2}} \right)$$

$$(7)$$

$$=\frac{\sum_{i}(x_{i}-\bar{x})^{2}\sigma_{\epsilon}^{2}}{\left[\sum_{i}(x_{i}-\bar{x})^{2}\right]^{2}}$$
(8)

$$=\frac{\sigma^2}{\sum_i (x_i - \bar{x})^2}\tag{9}$$

15 Similar to Eq. (6), we have

$$\sigma^2(K_{dp}) = \frac{\sigma^2(\psi_{dp})}{\frac{1}{3}\Delta r^2 \left[n(n-1)(n+1)\right]}. \quad \#$$
(10)

[response] The Gaussian mixture model is widely used in signal processing, but may be new in atmospheric science. To interpret this model, we may think the mixture model is that the data looks multimodal, for example, a raindrop size distribution (DSD) with multiple peaks. Trying to fit a multimodal DSD with a unimodal model will lead to poor fitting. An obvious way to
20 model a multimodal DSD would be to assume that it is generated by multiple unimodal DSD. In signal processing, a commonly used distribution is the Gaussian distribution. Therefore, modeling multimodal data as a mixture of many unimodel Gaussian distributions makes intuitive sense. We have added more words about Gaussian mixture model.

[changes] p.4, ln.25: Intuitively, it is used to model the multimodal data, with each Gaussian component corresponding to a subpopulation of the data.

**25 **1.2.** I think, that it is important to point out that Kdp is calculated from a filtered (estimated) differential phase and not directly from its moment-based measurements. To distinguish the three, one may use psi, fi, k symbols.**

[response] It is true that we need to derive  $\phi_{dp}$  from the raw data  $\psi_{dp}$  before estimating  $K_{dp}$ , since the X-band radar is affected by the backscattering differential phase  $\delta_{co}$ . In fact, the Gaussian mixture method analyzes the raw  $\psi_{dp}$  to calculate

the mean  $\psi_{dp}$  profile, and then remove  $\delta_{co}$  to obtain the  $\phi_{dp}$ . In the revision, we have corrected the notation problems by denoting the data as the  $\psi_{dp}$  before  $\delta_{co}$  elimination and as the  $\phi_{dp}$  after it.

[changes] We have made a number of changes:

- p.7, ln.20–30: From the chart of LR in Fig. 1.a, we can see that after the radar measurements are collected, the  $\psi_{dp}$  is unfolded, and then the clutter is removed. After these corrections, an iterative filtering method is applied to the  $\psi_{dp}$  profile. An adaptive method is finally used to estimate the  $K_{dp}$  profile according to the values of  $Z_H$ . The Gaussian mixture model, on the other hand, processes  $\psi_{dp}$  differently. First of all, the clutter is masked out according to the thresholds of  $Z_H$  and the variation of  $\psi_{dp}$ . Secondly, the range r and  $\psi_{dp}$  are fitted into a Gaussian mixture to yield the joint PDF, while the mean  $\psi_{dp}$  and the  $\psi_{dp}$  variance are obtained by taking the first raw and second central moments of the conditional PDF of  $\psi_{dp}$  given r. Thirdly,
- 10 some specific clusters in the Gaussian mixture PDF are adjusted to solve the problems of ambiguous  $\psi_{dp}$  and backscattering differential phase shift  $\delta_{co}$  in order to derive the PDF of  $\phi_{dp}$ . Fourthly, a raw  $K_{dp}$  profile is calculated from the first derivative of the expected values of  $\phi_{dp}$ , and the associated variances are obtained via a Taylor series expansion. Finally, the raw  $K_{dp}$  profile is smoothed, and consequently, the variances are reduced. In addition, new  $\phi_{dp}$  with lower variances can be re-constructed from the  $K_{dp}$  estimates.

**15 1.3.** The authors should emphasize that the main advantage of their proposed method is in providing the estimation variance for the Kdp and not is providing better estimates of Kdp. This is evident in the long-term evaluation using rain gauge data.**

[response] This is absolutely right that the Gaussian mixture method has the advantage that it provides the variance of  $K_{dp}$  together with the mean  $K_{dp}$ . Since the  $K_{dp}$  variance is nonconstant, it leads to the variability in the  $K_{dp}$  error characteristics.

- 20 Furthermore, the method yields the statistical uncertainty of  $K_{dp}$ , which is often missed in the existing methods. We can then use the uncertainty of  $K_{dp}$  to calculate the uncertainty of  $Z_H$  and  $Z_{DR}$  via the attenuation correction, and the uncertainty of R via the  $R-K_{dp}$  relation. These uncertainties are useful for studying the streamflow trends in the hydrological model. It is true that our rain rate estimates are not optimized for the MZZU radar, since we did not derive the  $R-K_{dp}$  relation in the paper. For the rain rate estimation, one can refer to some advanced studies, such as the IFloodS (Chen et al., 2017a) and MC3E
- 25 (Giangrande et al., 2014) campaigns.

[changes] We have made a number of changes:

p.2, ln. 34–p.3, ln.3: It is found that  $\sigma^2(K_{dp})$  is closely related to the square of the first derivative of  $K_{dp}$  and  $\sigma^2(\Phi_{dp})$ , while large  $\sigma^2(K_{dp})$  is associated with high variation of  $K_{dp}$  estimates. When compared to the existing methods, our method considers the joint probability density function of the data as the non-linear Gaussian mixture, leading to better performance

30 for the multimodal data. The  $K_{dp}$  variance can be used to calculate the variances of  $Z_H$ ,  $Z_{DR}$  and rain rate, and to study the streamflow trends in the hydrological model.

p.17, ln. 20–25: It is clear that the rain rates based on the GMM  $K_{dp}$  have a moderate consistency with the rain gauge data. To improve the results, some advanced rain rate algorithms can be considered, such as the rain-ice separation technique in the IFloodS campaign (Chen et al., 2017a) and the radar-gauge comparison method in the MC3E campaign (Giangrande et al.,

2014). Nevertheless, the GMM has the advantage over the existing methods, since it can yield the variance of  $K_{dp}$ . Furthermore,

the variance of R can also be obtained by the mean  $K_{dp}$  and the  $K_{dp}$  variance via the  $R-K_{dp}$  relation, leading to the variability in the error characteristics of R. Thus, the variances can be used to study the streamflow trends in the hydrological model.

**1.4.** I suggest that the authors improve the quality of the figures: some lettering is not legible, the inter-panel space could be reduced, etc.**

5 [response] We have improved the figures according to the suggestion. [changes] Figures 1, 2, 4, 7, 8 and 9.

**References**

Bringi, V. and Chandrasekar, V.: Polarimetric Doppler weather radar: Principles and applications, Cambridge Univ Pr, 2001.

- Chandrasekar, V., Chen, H., and Philips, B.: Principles of High-Resolution Radar Network for Hazard Mitigation and Disaster Management in an Urban Environment, Journal of the Meteorological Society of Japan. Ser. II, advpub, 2018.
- 5 Chen, H., Chandrasekar, V., and Bechini, R.: An Improved Dual-Polarization Radar Rainfall Algorithm (DROPS2.0): Application in NASA IFloodS Field Campaign, Journal of Hydrometeorology, 18, 917–937, 2017a.
  - Chen, H., Lim, S., Chandrasekar, V., and Jang, B.-J.: Urban Hydrological Applications of Dual-Polarization X-Band Radar: Case Study in Korea, Journal of Hydrologic Engineering, 22, E5016 001, 2017b.

Cifelli, R., Chandrasekar, V., Chen, H., and Johnson, L. E.: High resolution radar quantitative precipitation estimation in the San Francisco

- 10 Bay area: Rainfall monitoring for the urban environment, Journal of the Meteorological Society of Japan. Ser. II, 96, 141–155, 2018.
  - Giangrande, S. E., Collis, S., Theisen, A. K., and Tokay, A.: Precipitation Estimation from the ARM Distributed Radar Network during the MC3E Campaign, Journal of Applied Meteorology and Climatology, 53, 2130–2147, 2014.

---

## Author Comment (AC2) · 19 Jul 2019

Please find the responses in the supplement and the revised paper in the authors' general comments.

Please also note the supplement to this comment:
https://www.atmos-meas-tech-discuss.net/amt-2019-189/amt-2019-189-AC2-supplement.pdf

---

## Author Comment (AC3) · 19 Jul 2019

**A Gaussian Mixture Method for Specific Differential Phase Retrieval at X-band Frequency**

Guang Wen[1], Neil I. Fox[1,*], and Patrick S. Market[1]

[1]School of Natural Resources, University of Missouri, 332 ABNR Building, Columbia, Missouri, USA, 65201

**Correspondence:** Neil I. Fox (foxn@missouri.edu)

We would like to express our sincere thanks to the Editor for the efficient review management, and to the anonymous reviewers for their valuable comments and suggestions. We have addressed all the reviewers' comments point by point in the revision.

Review #3

**1  *General comments**

*1.1. It is not easy to provide a convincing validation of Kdp estimation, but some suggestions to improve it are provided below (note than pages 2-14 describes the method, but only pages 14-16 are related to validation). More difficult is to get an ultimate proof of the effectiveness of the method, but only a set of clues.*

[response] We sincerely appreciate the reviewer's efforts to give suggestions for the validation of the Gaussian mixture method (GMM). We are also aware that the papers in atmospheric science often take 30% for the method description and 70% for the application and validation. Due to the length of the paper, we have to separate the materials into multiple papers, so this paper is focused on the description of the Gaussian mixture method. We have also validated the method with case study and statistical analysis using the MZZU radar data. It is noted that our study is not shown in a single paper, but in a series of papers.

*1.2. The paper is correctly structured, but needs some improvement in order to be accepted for publication. The acceptance of the method will be depend also on practical aspects, such as the computational effort and robustness that, unfortunately are not considered in this manuscript.*

[response] We are very glad that the reviewer pointed out the weakness of the paper, and we try hard to improve the paper to meet the AMT standard. For the practical aspects including the computational effort and robustness, we will address these issues in comment 2.2.

**2  *Major comments**

*2.1. There is a problem in the use of symbols that affect the clarity of the manuscript. Kdp is estimated from the range derivative of differential propagation phase $\Phi_{dp}$, whereas the differential phase (usually indicated with $\Psi_{dp}$)that includes the differential phase upon backscattering, that is not negligible at X band) is measured by radar. Moreover, in a paper*

*about estimation methods, the estimated variable should be clearly distinct from the intrinsic variable. Usually, estimated variables are indicated with a hat on top. Adopting these standard symbols (or another set of symbols authors could prefer) would increase the readability of the manuscript.*

[responses] It is absolutely true that the notations were ambiguous in the first manuscript. We have fixed this problem by denoting the data as $\Psi_{dp}$ before the backscattering differential phase shift ($\delta_{co}$) elimination and as $\Phi_{dp}$ after it.

[changes] We have made a number of changes through the paper, for example:

p.7, ln.20–30: From the chart of LR in Fig. 1.a, we can see that after the radar measurements are collected, the $\Psi_{dp}$ is unfolded, and then the clutter is removed. After these corrections, an iterative filtering method is applied to the $\Psi_{dp}$ profile. An adaptive method is finally used to estimate the $K_{dp}$ profile according to the values of $Z_H$. The Gaussian mixture model, on the other hand, processes $\Psi_{dp}$ differently. First of all, the clutter is masked out according to the thresholds of $Z_H$ and the variation of $\Psi_{dp}$. Secondly, the range $r$ and $\Psi_{dp}$ are fitted into a Gaussian mixture to yield the joint PDF, while the $\Psi_{dp}$ mean and the $\Psi_{dp}$ variance are obtained by taking the first raw and second central moments of the conditional PDF of $\Psi_{dp}$ given $r$. Thirdly, some specific clusters in the Gaussian mixture PDF are adjusted to solve the problems of ambiguous $\Psi_{dp}$ and backscattering differential phase shift $\delta_{co}$ in order to derive the PDF of $\Phi_{dp}$. Fourthly, a raw $K_{dp}$ profile is calculated from the first derivative of the expected values of $\Phi_{dp}$, and the associated variances are obtained via a Taylor series expansion. Finally, the raw $K_{dp}$ profile is smoothed, and consequently, the variances are reduced. In addition, new $\Phi_{dp}$ with lower variances can be re-constructed from the $K_{dp}$ estimates.

**2.2. Aspects related to computational cost are neglected. Some figures about the computational efficiency of the method need to be provided.**

[responses] It is true that the computational time is important for the real-time application of the GMM. However, the time may vary for the PPIs. For example, the highest PPI in a volume takes less time than the lowest PPI, since it contains fewer points. It is also true that GMM takes more time than the linear regression method (LR). For the PPI data used in section 5, the LR takes about 1.47 seconds for phase unfolding on the PC, 0.458 seconds for $\Psi_{dp}$ smoothing and 0.109 seconds for regression-based $K_{dp}$ estimation. In total, the LR takes about 2.037 seconds if we ignore the time used for the inputs and outputs. On the other hand, the GMM uses 2.99 seconds for data masking, 2.348 seconds for $\Psi_{dp}$ density estimation, 0.73 seconds for $\Psi_{dp}$ unfolding and $\delta_{co}$ elimination and 0.98 seconds for $K_{dp}$ estimation. In total, we need 7.058 seconds for this case. If we skip the data masking process, we need about 4.068 seconds, about twice than the LR. Nevertheless, the GMM can obtain more information from the radar data than the LR.

We intend to make our codes available to the community soon, so the readers may apply the algorithm to their radar and see the computational time.

[changes] We have made a number of changes:

p.16, ln. 26–29: Moreover, the computational time is crucial for the real-time application of the $K_{dp}$ retrieval algorithms. For the data in Fig. 8, the GMM takes about 7.058/4.068 seconds to process the $K_{dp}$ with/without the data masking, whereas the LR reduces the time to about 2.037 seconds. It indicates that the LR has the advantages of simplicity and efficiency. Nevertheless, the GMM can obtain more information from the radar data, which is useful for the model studies.

**2.3.** *Introduction is quite confusing. It recall several Kdp estimation methods (some of them are missing, such as Vulpiani et. al. 2012, https://doi.org/10.1175/JAMC-D-10- 05024.1), but mixes general Kdp estimation methods with other methods based on self-consistency of dual polarization measurements that are valid strictly in rain.*

[responses] It is true that we missed the important paper about the $K_{dp}$ estimation for the C-band radars. We have cited the paper in the manuscript. Although the self-consistency relations may just be valid for rain, we respect the contributions to the X-band radars by the researchers who applied this technique to the $K_{dp}$ processing. Thus, we still cite these papers in this revision.

[changes] We have made a number of changes:

p.2, ln.11–12: Under the complex terrain, the $K_{dp}$ retrieval algorithm needs to be modified to obtain the accurate rainfall rate. Vulpiani et al. (2012) has opened new scenarios for the operational $K_{dp}$ processing in the Italian C-band radar network.

p.19, ln.3–4: In the future study, the algorithm will also be extended to other frequencies, such as C-band (Vulpiani et al., 2012; May et al., 1999) and S-band (Bringi and Chandrasekar, 2001).

**2.4.** *One of the advantages of GMM is that it can provide the variance associated the Kdp estimate. I think the authors should explain why such result is important and how it can be used. Other methods, such as Kalman filter can provide the variance of the estimate, but even linear regression can provide a standard error as the measure of the goodness of fitting.*

[responses] Indeed, the Gaussian mixture method has the advantage that it provides the variance of $K_{dp}$ together with the mean $K_{dp}$. Since the $K_{dp}$ variance is nonconstant, it leads to the variability in the $K_{dp}$ error characteristics. We can then use the variance of $K_{dp}$ to calculate the uncertainty of $Z_H$ and $Z_{DR}$ via the attenuation correction, and the uncertainty of $R$ via the $R$–$K_{dp}$ relation. These uncertainties are useful for studying the streamflow trends in the hydrological model.

The Kalman filter method is also an excellent method for the $K_{dp}$ estimation, since it can simultaneously obtain the $K_{dp}$, the attenuation-corrected $Z_H$, the attenuation-corrected $Z_{DR}$ and $\delta_{co}$. The method is then adapted to various environmental conditions by considering $\Psi_{dp}$ only. From Schneebeli et al. (2014), we can see that the primary difference is that the Kalman filter method assumes the error covariance function follows a linear Gaussian distribution, whereas the GMM considers the joint PDF of $r$ and $\Psi_{dp}$ as a non-linear Gaussian mixture. Thus, the GMM may have better performance when the data are multimodal. Moreover, the Kalman filter method derive a priori from the measured DSD, leading to some constraints on the particle types. In contrast, the GMM fits the data to obtain the random errors of $\Psi_{dp}$ data.

We will discuss the linear regression method in comment 2.5.

[changes] We have made a number of changes:

p.2, ln.34–p.3, ln.3: It is found that $\sigma^2(K_{dp})$ is closely related to the square of the first derivative of $K_{dp}$ and $\sigma^2(\Phi_{dp})$, while large $\sigma^2(K_{dp})$ is associated with high variation of $K_{dp}$ estimates. When compared to the existing methods, our method considers the joint probability density function of the data as the non-linear Gaussian mixture, leading to better performance for the multimodal data. The $K_{dp}$ variance can be used to calculate the variances of $Z_H$, $Z_{DR}$ and rain rate, and to study the streamflow trends in the hydrological model.

p.2, ln.26–28: It is noticeable that the Kalman filter method minimizes the Gaussian error function to obtain the mean profile of $K_{dp}$. It gives a significant improvement on the $K_{dp}$ mean, particularly in the small-scale structure with high peaks.

**2.5.** *The manuscript states that $\sigma(\Phi_{dp})$ is constant. However, notoriously $\sigma(\Phi_{dp})$ varies depending on SNR, width of Doppler spectrum and copolar correlation coefficients that are not constant along the range.*

[responses] We calculated the $\sigma(\Phi_{dp})$ based on the work in Williams and May (2008) and Bringi and Chandrasekar (2001). For the MZZU radar, it is equal to $2.61°$ by assuming 32 pulses, 1 m s$^{-1}$ for Doppler spectrum and 0.98 for $\rho_{hv}$. It is true that the SNR and Doppler spectrum may vary along the range, leading a variable $\sigma(\Phi_{dp})$. However, in weather radar signal processing, the estimates of $\Phi_{dp}$ are obtained by averaging a number of continuous samples of the time-series data, yielding a stable $\sigma(\Phi_{dp})$. Sachidananda and Zrnic (1986) showed that $\sigma(\Phi_{dp})$ is only a few degree. In theoretical work, we often assume that $\sigma(\Phi_{dp})$ is constant, since the signal processor normally does not output $\sigma(\Phi_{dp})$.

[changes] We have made a number of changes.

p.2, ln.30–31: The $K_{dp}$ variance is often inherited from the $\Phi_{dp}$ variance $\left(\sigma^2(\Phi_{dp})\right)$ leading to large relative errors for low $K_{dp}$ with a fixed path length.

p.10, ln.12: whereas these errors $\sigma(\Psi_{dp})$ are often considered as small and stable values in LR.

p.13, ln.13: indicating $\sigma^2(K_{dp})$ is also stable in LR.

**3    *Minor comments**

**3.1.** *Page 1: Line 10, raingauge data used for validation are relative to two years and 3 months, not three years.*

[changes] It has been changed to

p.1, ln.10: Furthermore, the performance is quantitatively assessed by two-year radar-gauge data, and the results are compared to linear regression model.

**3.2.** *Page 1: Line 15, please use different symbols for the measure differential phase shift and its component related to propagation.*

[responses] As addressed in comment 2.1, we have fixed the notation problem in this revision.

**3.3.** *Page 2: Line 5-6, "Therefore...". This is something to be demonstrated, it is not a consequence of the previous statement.*

[changes] If this comment means p.3, l.5–6, we have removed "Therefore".

**3.4.** *Page 5: Line 25-30: Is there an implication of this sentence for Kdp estimation ?*

[responses] No, these sentences discussed the speeds for fitting a Gaussian mixture using the E-M algorithm or a typical regression problem under the framework of the Gaussian mixture model. We have showed the speed of the GMM for $K_{dp}$ estimation via an example in section 5.1.

[changes] We have made a number of changes:

p.16, ln. 26–29: Moreover, the computational time is crucial for the real-time application of the $K_{dp}$ retrieval algorithms. For the data in Fig. 8, the GMM takes about 7.058/4.068 seconds to process the $K_{dp}$ with/without the data masking, whereas the LR reduces the time to about 2.037 seconds. It indicates that the LR has the advantages of simplicity and efficiency. Nevertheless, the GMM can obtain more information from the radar data, which is useful for the model studies.

**3.5.** *Page 6: Line 10. What is "the maximum detectable range"? is it the unambiguous range determined by the selected PRF ?*

[responses] It is true that we mean the unambiguous range here.

[changes] p.6, ln.17–18: The maximum unambiguous range of the MZZU radar is 94.64 km with a resolution of 260 m in range and $1°$ in azimuth.

**3.6.** *Page 6: Line 10-25: are the height above the ground or ASL ? for the elevation of $0.8°$, at 4.4 km from radar the height of radar beam is 314.6 m. Is it correct ?*

[responses] It is ASL. Yes, it is true. We consider the altitude of our radar tower.

[changes] It has been changed to

p.6, ln.24–26: The first elevations at Bradford and Sanborn may be affected by ground clutter, since the radar beams are very close to the ground, with heights of 314.6 m and 336.9 m ASL, respectively, including the radar tower.

**3.7.** *Page 8: Line 1-2: There is something odds in the sentence. Are clutter-contaminated echoes well identified or not ?*

[changes] It has been changed to

p.8, ln.10–11: The clutter may be incorrectly identified in the regions of high reflectivity or for the echoes mixed by weather and clutter.

**3.8.** *Page 8: lines 9-11. What is the need of eliminate hail contamination ? Is not Kdp computed in hail ?*

[responses] Since melting hails or raindrops with ice cores make a significant contribution to the variation of $\Phi_{dp}$, we need to increase the thresholds for data masking in the regions of a mixture of rain and hail.

[changes] It has been changed to

p.8, ln.18–19: To reduce the mis-classification in the hail regions, the thresholds are increased for higher $Z_H$, resulting in $\sigma(\Psi_{dp})/\sigma(r) < 47.9° \text{ km}^{-1}$ and $\sigma(\Psi_{dp}) < 6.3°$.

**3.9.** *Page 8: Lines 18-32: I recommend to use degrees consistently in the manuscript and add a legend on x-axis (this is valid for all the figures showing profiles). Given the range of unambiguous differential phase, it seems that MZZU features the alternate polarization scheme. About the jump at the beginning of the profile in Fig. 3b: to be interpreted as $\delta_{co}$ there should a peak of Zdr. Is this the case ?*

[responses] We have updated Figs. 3, 4 and 6. The cut-off in the range is due to the strong attenuation at X-band. Indeed, there is a peak in $Z_{DR}$ in Fig. 1, however, the GMM does not rely on $Z_{DR}$ for the $\delta_{co}$ estimation. So, we would not show $Z_{DR}$ in this revision.

**3.10.** *Page 9. Lines 1-2: negative Kdp can be due also to non uniform beam filling (Ryzkhov 2007 https://doi.org/10.1175/JTECH2003. which is a further source of error in computing Kdp (see manuscript at pag 15).*

[responses] The non-uniform beam filling may affect the error characteristics of $K_{dp}$, but we think the particle types primarily determine the sign of $K_{dp}$ here. We have given a reference about it (Marzano et al., 2010). The non-uniform beam filling is beyond the scope of this paper, so we would not give any reference about it. The mis-matches between radar and rain gauge are likely produced by the wet-radome attenuation. The readers may refer to the QPE papers for further information.

[changes] We have made a number of changes:

[Figure]

**Figure 1.** $Z_{DR}$ for Fig. 4.b

p.9, ln.8–9: It rises quickly for horizontally-oriented anisotropic scatterers, and conversely, it falls steadily for vertically-oriented particles (Marzano et al., 2010).

**3.11.** *Page 10. Again, please be sure that the interpretation of differential phase upon backscattering is correct.*

[responses] We think our interpretation of $\delta_{co}$ is basically correct. To further clarify it, we have given the definition of $\delta_{co}$.

5    [changes] We have made a number of changes:

p.11, ln.22–24: which is defined as the phase difference between the horizontal and vertical polarizations upon the backscattering of the particles in a radar resolution volume.

**3.12.** *Page 11. Line 29. Section explains how unfolding works. Sometimes there are false alarms in unfolding. Could you provide the rate of correct unfolding for the dataset used?*

10    [responses] The false alarms do not occur very frequently, and we have not yet found any clear evidence of errors in unfolding in our two-year dataset. In some cases, the clutter may result in a false alarm, so we assume that the first cluster should be less than a certain threshold ($< 90°$). Even if the false alarm may occur, the $K_{dp}$ smoothing removes the spike effect, producing the correct $K_{dp}$ means.

**3.13.** *Page 13, line 22-23. Could authors please provide more details on how the weights are derived?*

15    [responses] We generated the FIR filter using a window method. We have then given some details of the parameters in the method.

[changes] We have made a number of changes.

p.13, ln.31–33: Figure 7 shows the time responses of the FIR with the cutoff frequency of 0.053 and the Gaussian window of 28, which yield the best performance for the MZZU radar.

20    p.14, ln.1–4: In this study, we gradually increase the order number to calculate the difference between the $K_{dp}$ profiles obtained by the FIR filters with two adjacent order numbers. The optimal order of the FIR filter is then set when the relative

square error of the two $K_{dp}$ is below 0.001. For profiles with sufficiently large data points, the order number is between 29 and 33 for the MZZU radar.

**3.14.** *Page 14: Line 23. Please provide information about how hourly rain is obtained from instantaneous radar measurements.*

[changes] We have made a number of changes:

p.15, ln.9–11: It is noted that the radar collects instantaneous measurements every 4–5 minutes, whereas RGs obtain the precipitation accumulations over 60 minutes. Therefore, it is necessary to average 12–15 consecutive radar scans to derive the hourly rain amounts.

**3.15.** *Page 15: Line 13-24. The presence of too many missing data for LR indicates a clear advantage for GMM. However, it is not clear what is the cause of these missing data. Surprisingly is the presence of Kdp estimates beyond the edge of differential phase shift rays, especially for LR. Is it the effect of smoothing and/or extrapolation?*

[responses] If we look at the raw data of $Z_H$ and $\Phi_{dp}$, we can identify a number of missing data. The regions of the missing data are extended due to the smoothing in the linear regression method. In contrast, the PDF of $\Phi_{dp}$ in the GMM can still produce the $K_{dp}$ means at these regions, giving better $K_{dp}$ estimates than the LR. The extended $K_{dp}$ values at the edges are also due to the smoothing. We fill zero values at the edges by half of the window for the FIR filter. The resulting PPI will then present zero values at the edge.

[changes] We have made a number of changes:

p.16, ln.2–4: Nevertheless, LR (Fig. 9.b) produces continuous $K_{dp}$ by $\Phi_{dp}$ unfolding and linear interpolation according to the trends of the profiles, but some missing data still exist within the storm, due to low signal-to-noise ratio.

**3.16.** *Page 15: Line 25. Notoriously, in rain, Kdp, Zh, and Zdr exhibit a self-consistency property. I recommend authors to exploit this property to validate their method to see whether the Kdp estimate is consistent with Zh and Zdr measurements.*

[responses] That is an excellent idea. We have also considered the attenuation correction in our work. However, we think this paper is very long, and it has all the components for a research article. We will include it in our future work.

Here are some examples.

**3.17.** *Page 32: Figure 7: Please use larger font size.*

[changes] We have improved the layout and increased the font size.

**3.18.** *Page 34: A Kdp rain algorithm is supposed to work well (at least better than a R(Z) algorithm) for high rainfall rate. Instead, there are evident underestimates for high rainfall rates that likely affect RMSE. How about LR: does it yield the same behavior?*

[responses] Yes, the LR yields a similar behaviour. This underestimation is likely due to the wet radome attenuation. Our study shows that $R(Z)$ is much worse than $R(K_{dp})$, giving a correlation coefficient of about 60%. In the revision, we have provided the radar-gauge comparison figures for the LR (see Fig. 10).

**3.19.** *Page 36: NB is not in mm.*

[responses] It is true. The NB should be unitless.

[Figure]

**Figure 2.** Self-consistency for $Z_H$, $Z_{DR}$ and $K_{dp}$.

**References**

Bringi, V. and Chandrasekar, V.: Polarimetric Doppler weather radar: Principles and applications, Cambridge Univ Pr, 2001.

Marzano, F. S., Botta, G., and Montopoli, M.: Iterative Bayesian Retrieval of Hydrometeor Content From X-Band Polarimetric Weather Radar, Geoscience and Remote Sensing, IEEE Transactions on, 48, 3059–3074, 2010.

May, P. T., Keenan, T. D., Zrnić, D. S., Carey, L. D., and Rutledge, S. A.: Polarimetric Radar Measurements of Tropical Rain at a 5-cm Wavelength, Journal of Applied Meteorology, 38, 750–765, 1999.

5   Sachidananda, M. and Zrnic, D. S.: Differential propagation phase shift and rainfall rate estimation, Radio Sci., 21, 235–247, https://doi.org/10.1029/RS021i002p00235, http://dx.doi.org/10.1029/RS021i002p00235, 1986.

Schneebeli, M., Grazioli, J., and Berne, A.: Improved estimation of the specific differential phase shift using a compilation of Kalman filter ensembles, IEEE Transactions on Geoscience and Remote Sensing, 52, 5137–5149, 2014.

Vulpiani, G., Montopoli, M., Passeri, L. D., Gioia, A. G., Giordano, P., and Marzano, F. S.: On the Use of Dual-Polarized C-Band Radar for
10   Operational Rainfall Retrieval in Mountainous Areas, Journal of Applied Meteorology and Climatology, 51, 405–425, 2012.

Williams, C. R. and May, P. T.: Uncertainties in Profiler and Polarimetric DSD Estimates and Their Relation to Rainfall Uncertainties, Journal of Atmospheric and Oceanic Technology, 25, 1881–1887, 2008.